# A senataxin-associated exonuclease SAN1 is required for resistance to DNA interstrand cross-links

Alex M. Andrews[1], Heather J. McCartney[1], Tim M. Errington [2,4], Alan D. D'Andrea[3] & Ian G. Macara[1]

Interstrand DNA cross-links (ICLs) block both replication and transcription, and are commonly repaired by the Fanconi anemia (FA) pathway. However, FA-independent repair mechanisms of ICLs remain poorly understood. Here we report a previously uncharacterized protein, SAN1, as a 5′ exonuclease that acts independently of the FA pathway in response to ICLs. Deletion of SAN1 in HeLa cells and mouse embryonic fibroblasts causes sensitivity to ICLs, which is prevented by re-expression of wild type but not nuclease-dead SAN1. SAN1 deletion causes DNA damage and radial chromosome formation following treatment with Mitomycin C, phenocopying defects in the FA pathway. However, SAN1 deletion is not epistatic with FANCD2, a core FA pathway component. Unexpectedly, SAN1 binds to Senataxin (SETX), an RNA/DNA helicase that resolves R-loops. SAN1-SETX binding is increased by ICLs, and is required to prevent cross-link sensitivity. We propose that SAN1 functions with SETX in a pathway necessary for resistance to ICLs.

[1] Department of Cell and Developmental Biology, Vanderbilt University School of Medicine, Nashville, TN 37215, USA. [2] Department of Microbiology, Immunology and Cancer Biology, University of Virginia, Charlottesville, VA 22908, USA. [3] Dana-Farber Cancer Institute, Harvard Medical School, Boston, MA 02215, USA. [4] Present address: Center for Open Science, 210 Ridge McIntire Road, Suite 500, Charlottesville, VA 22903, USA. These authors contributed equally: Alex M. Andrews, Heather J. McCartney. Correspondence and requests for materials should be addressed to I.G.M. (email: ian.g.macara@vanderbilt.edu)

I nterstrand cross-links (ICLs) are a toxic form of damage that disrupts transcription and replication by covalently joining complementary DNA strands. ICL repair requires the collective involvement of nucleotide excision repair (NER), translesion synthesis (TLS), and homologous recombination (HR), the integration of which is still not fully understood. Although many components of these pathways are conserved between yeast and higher organisms, animals have evolved an additional network of >20 proteins specialized for ICL repair, called the Fanconi anemia

(FA) pathway[1,2]. Generally, ICL repair occurs in S phase when replication forks collide with the lesion, which activates the FA pathway[3,4], although a replication-independent pathway involving transcription-coupled repair (TCR) has also been proposed[5]. Mono-ubiquitylation of the FANCI-FANCD2 (ID2) heterodimer leads to recruitment of multiple nucleases that control nucleolytic incision and ICL unhooking[6], including the endonucleases XPF (FANCQ), which forms an XPF-ERCC1 heterodimer, and SLX1, with additional nucleases such as FAN1, SNM1A, and MUS81

contributing independently of the FA pathway[7,8]. XPF-ERCC1, which is involved in nucleotide excision repair, is recruited to perform the unhooking incisions[9,10], but under some circumstances it only performs the 3′ incision, and another nuclease would be responsible for the 5′ incision. The identity of this nuclease remains ambiguous but SLX1 is one candidate[11]. FAN1, a nuclease that interacts with FANCD2, can digest recessed 5′ DNA ends and cleave 4 nt 3′ to an ICL[12], but is not required for unhooking in *Xenopus* extracts and its function in ICL repair remains unclear[9]. Another nuclease, SNM1A, has no known function in incision, but may participate in repair by digesting past the ICL[7]. It remains unclear whether a single nuclease is responsible for the 5′ incision, or if several nucleases act redundantly to complete this process.

ICL repair can also be triggered by stalling of transcription complexes at lesions during other periods of the cell cycle, including G1[5]. One consequence of transcriptional stalling is the formation of R-loops, which consist of a RNA–DNA hybrid plus the looped single-stranded coding strand of the DNA[13,14]. R-loops form naturally during transcription at promoters of genes with a high GC content and at termination regions of genes[15,16]. Persistent R-loops can impede replication and be processed into double stranded breaks (DSBs), leading to genomic instability. ICLs between RNA and DNA strands might also occur at these structures[17], although to date there is no direct evidence for their existence. R-loops can be resolved by an endonuclease, RNase H, or by an RNA/DNA helicase, senataxin (SETX), and if they persist can be aberrantly processed into DSBs by the NER endonucleases XPF and XPG[14]. Interestingly, R-loop resolution has recently been linked to the FA pathway[17,18], and to BRCA1[19], a protein essential for HR and resistance to ICLs.

Our knowledge of the DNA repair machinery remains incomplete, and additional nucleases might exist for removal of specific lesions. We identified an uncharacterized protein that contains an N-terminal domain closely related to the FEN1 family of structure-specific nucleases. This protein is not a known component of any DNA repair complex, but we now report that it is a 5′-exonuclease for single-stranded (ss) DNA, and is required for the cellular response to ICLs. Disruption of the gene for this nuclease, *Fam120b*, has no effect on cell cycle but specifically increases sensitivity to cross-linking agents, and results in DNA damage and radial chromosome formation in the presence of Mitomycin C (MMC). However, it is not epistatic with components of the FA pathway including FANCD2 and XPF. Rather the nuclease binds to the RNA/DNA helicase SETX, and for this reason we propose the name SAN1 (Senataxin-associated nuclease 1). This interaction is required for SAN1 function. SAN1 cooperates with SETX in an FA-independent repair process to protect cells from ICLs. It is the first nuclease to be identified in the cellular response to ICLs since the discovery of FAN1, and will provide an important gateway to understand FA-independent mechanisms of DNA repair required for the resolution of ICLs.

## Results

**SAN1 is a 5′ exonuclease specific for single-stranded DNA.** The *Fam120b* gene product was originally described as a transcriptional co-activator of PPAR-gamma[20]. We noticed, however, that the N-terminal region of this protein is closely related to the FEN1 family of structure-specific nucleases, which are involved in DNA replication, recombination, and various DNA repair pathways. They possess distinct features conserved among family members, including a nuclease domain containing seven acidic residues essential for enzymatic activity[21]. These residues are conserved in SAN1 (Supplementary Figure 1a). Moreover, the Robetta server found a confident match between this N-terminal domain and the crystal structure of *A. fulgidus* FEN1, and the locations of the active site carboxylates correspond very closely with those of FEN1 (Fig. 1a, Supplementary Figure 1a, b). While absent from *Drosophila* or *C. elegans* genomes, SAN1 orthologs were identified in other insects and the lower Metazoa, including the sea anemone and hydra (Supplementary Figure 1c). In addition to the N-terminal nuclease-like domain, SAN1 also possesses a central region of repeated 12 residue motifs containing the sequence QEVPM (Supplementary Figure 1d), and a conserved C-terminal region. Different species contain different numbers of repeats, varying from 2 to 12.

Based on the concordance with FEN1, we asked if SAN1 has detectable nuclease activity. The protein was expressed in *E. coli* with a C-terminal Strep-tag and purified over Strep-Tactin beads, with stringent washes prior to elution with desthiobiotin (Fig. 1b). We used murine SAN1 (mSAN1) because it contains fewer central repeat motifs than the human protein. As a control for contaminating nucleases, we created a version in which the highly conserved Asp-90 residue in the putative active site (Fig. 1a) was mutated to Alanine (D90A), and was purified using an identical protocol.

To assess the potential nuclease activity of SAN1, we tested the ability of mSAN1 to cleave [32]P 5′-labeled, ssDNA, using 2 different 50 nt oligonucleotides (X1, X4, see Table 1 for sequences). After incubation with SAN1 WT or SAN1 D90A, reaction products were visualized by denaturing polyacrylamide gel electrophoresis and autoradiography. The wild type protein efficiently cleaved the 5′ end of the substrate (Fig. 1c). Importantly, the D90A mutant had no detectable activity, demonstrating that the observed activity of the WT protein did not result from bacterial contaminants. Moreover, the D90A

**Fig. 1** Identification and characterization of the SAN1 nuclease. **a** Modeling of conserved carboxylates in active site of SAN1 (light blue), using the Robetta server (http://robetta.bakerlab.org) and the *A. fulgidus* FEN1 structure (PDB 1RXW) (green) as template. Residue highlighted by red box is the aspartate mutated to make D90A. **b** Murine SAN1 was expressed with a C-terminal Strep tag in *E. coli* and purified over Strep-Tactin beads. Purified protein (0.2 μg) was analyzed by PAGE and stained with Coomassie. Arrow shows mSAN1 expected size of 100 kD. **c** Synthetic 50-mer oligos were 5′ labeled with [32]P and incubated with mSAN1 for 120 min. Products were separated by PAGE and [32]P-fragments were detected by autoradiography (see Table 1 for sequences). **d** 50 nt X4 was 5′ [32]P labeled and incubated with RPA as a positive control or increasing concentrations of SAN1 D90A (catalytically inactive). Samples were analyzed on a native gel and exposed to X-ray film. **e** Schematic of double-affinity Strep-FLAG tag purification for human SAN1 WT and SAN1 DA. **f** Silver stained fractions from the purification where "W" denotes Wash steps and "E" denotes Elution steps for the Strep and FLAG IPs. Arrow shows human SAN1 WT (expected size 150 kD) and asterisks show FLAG antibody heavy and light chains. **g** Top panel shows immunoblot of fractions from two-step purification of SAN1 where arrow shows SAN1 (expected size 150 kD), detected using mouse M2 anti-FLAG-antibody. Bottom panel shows corresponding filter spin nuclease assay. **h** X1 (50-mer ssDNA) was 5′ labeled with [32]P and incubated with SAN1 WT or the D90A mutant. Products were analyzed as in **c**. **i** X4 ssDNA or dsDNA X1 + X4 were 3′ [32]P labeled and incubated with WT or D90A SAN1. Products were analyzed as in **h**. **j** FLAG-tagged SAN1 WT or D90A was incubated with 5′ [32]P labeled splayed duplex, 3′ flap, or 5′ flap structures for 2 h at 37 °C. Products were processed as in **c**. **k** Using the filter spin assay, initial rates of 5′ [32]P-labeled X4 hydrolysis were measured at different substrate concentrations. Line was fitted using Prism software, assuming Michaelis-Menten kinetics

**Table 1 Oligos used in nuclease assays**

| | | | |
|---|---|---|---|
| X1 | 5′-TGGGTCAACGTGGGCAAAGATGTCCTAGCAATGTAATCGTCTATGACGTT-3′ | ssDNA | 50 nt |
| X4 | 5′-AACGTCATAGACGATTACATTGCTAGGACATCTTTGCCCACGTTGACCCA-3′ | ssDNA | 50 nt |
| a3 | 5′-CCTCGATCCTACCAACCAGATGACGCGCTGCTACGTGCTACCGGAAGTCG-3′ | Splayed arm, 3′ flap, 5′ flap, replication fork | 50 nt |
| b | 5′-CGACTTCCGGTAGCACGTAGCAGCGGCTCGCCACGAACTGCACTCTAGGC-3′ | Splayed arm, 3′ flap, 5′ flap, replication fork | 50 nt |
| c | 5′-GCCTAGAGTGCAGTTCGTGGCGAGC-3′ | Splayed arm, 3′ flap, 5′ flap, replication fork | 25 nt |
| d3 | 5′-CGTCATCTGGTTGGTAGGATCGAGG-3′ | Splayed arm, 3′ flap, 5′ flap, replication fork | 25 nt |
| X3 | 5′-TGGGTCAACGTGGGCAAAGATGTCC-3′ | ssDNA, first 25 nt of X1 | 25 nt |
| N1 | 5′-TGGGTCAACGTGGGCAAAGA-3′ | Nicked | 20 nt |
| N2 | 5′-ATGTAATCGTCTATGACGTT-3′ | Nicked | 20 nt |
| N3 | 5′-AACGTCATAGACGATTACATTCTTTGCCCACGTTGACCCA-3′ | Nicked | 40 nt |
| G1 | 5′-TGGGTCAACGTGGGCAAAG-3′ | Gapped | 19 nt |
| G2 | 5′-ATGTAATCGTCTATGACGTT-3′ | Gapped | 20 nt |
| G3 | 5′-AACGTCATAGACGATTACATTCTTTGCCCACGTTGACCCA-3′ | Gapped | 40 nt |
| X4 Biotin 5′ | 5′-Biotin-AACGTCATAGACGATTACATTGCTAGGACATCTTTGCCCACGTTGACCCA-3′ | X4—5′ Biotin | 50 nt |
| X1A | 5′-TGGGTCAACGTGGGCAAAGATTTTTTTTTTTTTTTTTTTTTGTCCTAGCAATGTAATCTG-3′ | Variation of X1—1st 20 nt of X1, 20 internal Ts, middle 20 nt of X1 | 60 nt |
| X1B | 5′-TTTTTTTTTTTTTTTTTTTTGGGTCAACGTGGGCAAAGATTTTTTTTTTTTTTTTTTTT-3′ | Variation of X1—20 Ts, 1st 20 nt of X1, 20 Ts | 60 nt |
| S3 | 5′-TGGGTCAACG-3′ | Ladder—1st 10 nt from X1 oligo | 10 nt |
| S4 | 5′-TGGGTCAACGTGGGC-3′ | Ladder—1st 15 nt from X1 oligo | 15 nt |
| S5 | 5′-TGGGTCAACGTGGGCAAAGA-3′ | Ladder—1st 20 nt from X1 oligo | 20 nt |
| S6 | 5′-TGGGTCAACGTGGGCAAAGATGTCC-3′ | Ladder—1st 25 nt from X1 oligo | 25 nt |
| S7 | 5′-TGGGTCAACGTGGGCAAAGATGTCCTAGCA-3′ | Ladder—1st 30 nt from X1 oligo | 30 nt |
| X0-1 | 5′-ACGCTGCCGAATTCTACCAGTGCCTTGCTAGGACATCTTTGCCCACCTGCAGGTTCACCC-3′ | Holliday Junction | 60 nt |
| X0-2 | 5′-GGGTGAACCTGCAGGTGGGCAAAGATGTCCATCTGTTGTAATCGTCAAGCTTTATGCCGT-3′ | Holliday Junction | 60 nt |
| X0-3 | 5′-ACGGCATAAAGCTTGACGATTACAACAGATCATGGAGCTGTCTAGAGGATCCGACTATCG-3′ | Holliday Junction | 60 nt |
| X0-4 | 5′-CGATAGTCGGATCCTCTAGACAGCTCCATGTAGCAAGGCACTGGTAGAATTCGGCAGCGT-3′ | Holliday Junction | 60 nt |

mutant could bind ssDNA, as detected by EMSA assay (Fig. 1d), demonstrating that it is correctly folded.

We also expressed human Strep$_2$-FLAG-tagged (ssf) SAN1 in HEK 293T cells and purified it by a double-affinity method (Fig. 1e). Lysate protein was bound to Strep-Tactin beads, washed and eluted with desthiobiotin, then bound to anti-FLAG M2 beads, and eluted with FLAG peptide. Nuclease activity was found only in the fractions containing SAN1 (Fig. 1f, g), strongly suggesting that the activity is intrinsic to this protein.

Interestingly, incubation of SAN1-ssf with 5′-$^{32}$P-labeled ssDNA substrate generated ~3–7 nucleotide fragments rather than single nucleotides (Fig. 1h, and Supplementary Figure 1f), properties reminiscent of FAN1, which also resects 5′ overhangs in steps of ~4 nts[22,23]. A 3′-labeled ssDNA oligonucleotide produced a ladder of longer fragments (Fig. 1i), suggesting that SAN1 activity is not processive. Additionally, the D90A mutation blocked all product formation, confirming that the observed nuclease activity is intrinsic to the WT protein (Fig. 1h, i, Supplementary Figure 1g). Collectively, these data demonstrate that SAN1 has specific 5′ exonuclease activity on ssDNA. Because the human protein was more stable than the murine protein expressed in bacteria, we used this preparation, purified over anti-FLAG M2 beads, for further studies. We next tested a variety of synthetic oligonucleotides resembling DNA replication intermediates, labeled at the 5′-end of each of the strands (Fig. 1j and Supplementary Figure S1e). Linear dsDNA with no 5′ overhang

was not cleaved. Only structures possessing a free 5′ ssDNA flap were substrates for SAN1 (Fig. 1j). Interestingly, a splayed arm structure appears to be important because cleavage of a 5′ flap is less efficient when the 3′ arm is double-stranded (Fig. 1j, lane 2 versus lane 4).

Using a filter spin nuclease assay we measured the nucleolytic rate of SAN1 to compare its activity to other FEN1 family members (Fig. 1k). From kinetic measurements we estimate a Km for ssDNA of ~0.46 μM, and kcat of 10.6 min$^{-1}$ (Fig. 1k), similar to the kinetic constants for other nucleases in the FEN1 family (e.g., for Exo1, kcat = 5.6 min$^{-1}$ with a ssDNA as substrate[24]. Kinetic measurements were taken within the linear phase of the reaction (Supplementary Figure 1h). A time course of SAN1 activity on a ssDNA substrate is also shown in Supplementary Figure 2b.

We next tested a variety of additional possible substrates for SAN1 including a 25 nt single-stranded ssDNA oligonucleotide (Fig. 2a). In contrast to the 50 nt ssDNA substrate, and despite displaying clear nuclease activity on the 25 nt 5′ flap of a splayed arm substrate (Fig. 1j), SAN1 displayed no activity towards the 25 nt ssDNA fragment (Fig. 2a). This result suggests that like other members of the FEN1 family, SAN1 recognizes structural elements in its substrates. Further supporting this possibility SAN1 can, albeit inefficiently, cleave a 5′ ssDNA overhang of only 20 nts when directly adjacent to a region of dsDNA (Fig. 2b). SAN1 displayed no activity towards nicked and gapped substrates

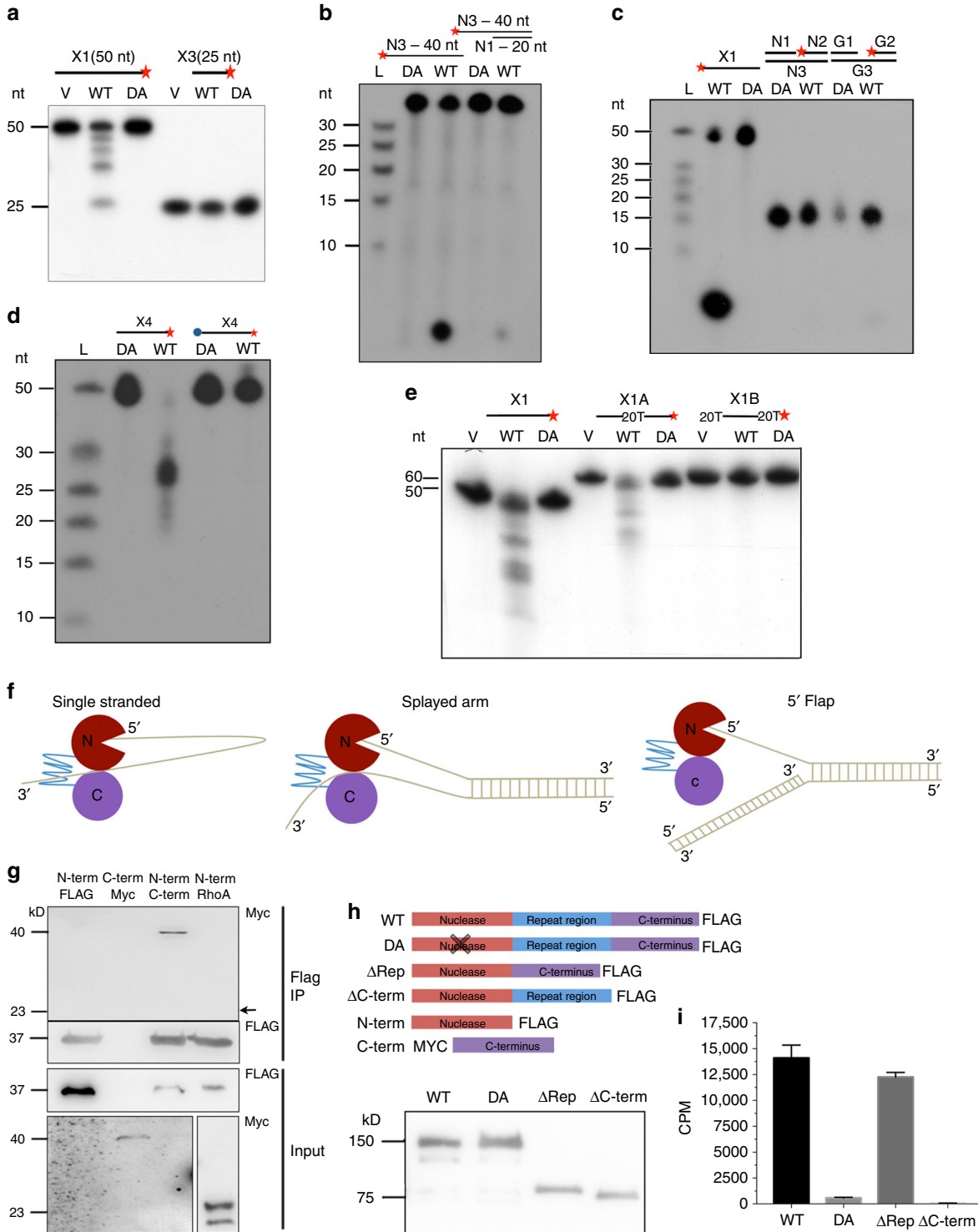

**Fig. 2** SAN1 acts as a 5′ exonuclease on single-stranded DNA substrates. Affinity-purified FLAG-SAN1 WT or -D90A from 293T cells incubated with **a** 50 nt ssDNA versus 25 nt ssDNA; **b** 40 nt ssDNA versus a 20 nt ssDNA 5′ overhang followed by 20 bp of dsDNA; **c** dsDNA oligos with an internal nick or gap; **d** 5′ biotinylated X4 and unbiotinylated X4; **e** variants of ssDNA oligo X1 with 20 Ts 5′ and 3′ to 20 nts of X1, or a tract of 20 Ts bounded by two 20 nt sections of the X1 sequence all for 2 h at 37 °C. DNA structures assayed in **b**, **c** were 5′ $^{32}$P labeled and structures in **a**, **d**, **e** were 3′ $^{32}$P labeled. Products were separated by PAGE and $^{32}$P fragments were detected by autoradiography. **f** Schematic of a model for SAN1 nuclease activity. SAN1 acts on ssDNA substrates by recognizing the free 5′ end of DNA and cleaving ~3 or ~7 nts from the 5′ end, in a non-processive manner. **g** The N-terminal FLAG-tagged nuclease domain was co-expressed in 293T cells with C-terminal Myc-tagged SAN1 C-terminus, or Myc-RhoA as a negative control. Lysates were precipitated with anti-FLAG M2 beads and analyzed by immunoblot. The Myc Input membrane was re-exposed for a longer time to detect Myc-SAN1 C-terminus. The SAN1 C-terminal domain but not RhoA is co-precipitated with the nuclease domain. **h** Schematic of SAN1 deletion mutants used in **f**, **g**, **h** followed by immunoblot of WT, DA, ΔRep, and ΔC-term proteins purified from 293T cells; and **i** tested for nuclease activity against 5′ $^{32}$P labeled ssDNA using the filter spin assay ($N = 2$, error bars show range)

(Fig. 2c), or Holliday junctions (HJs) (Supplementary Figure 2a). To determine if a free 5′ end was required for SAN1 activity we tested a 5′ biotinylated 50 nt oligonucleotide as substrate, which was not detectably cleaved (Fig. 2d), consistent with exonucleolytic activity.

Finally, we asked if SAN1 exhibited any substrate specificity between differing ssDNA oligonucleotides, based on our previous observation that a 3′ labeled ssDNA substrate produced a ladder

of products down to a minimum size of ~25 nt (Fig. 1i). Interestingly, inclusion of 20 dTs in the middle of a 60 nt substrate changed the pattern of SAN1 cleavage. Moreover, placement of 20 dTs at the beginning and end of the substrate completely blocked digestion, suggesting that thymidine tracts are not well recognized by SAN1 (Fig. 2e).

Together, these data reveal that SAN1 has a unique 5′ exonuclease activity that is non-processive, requires a free

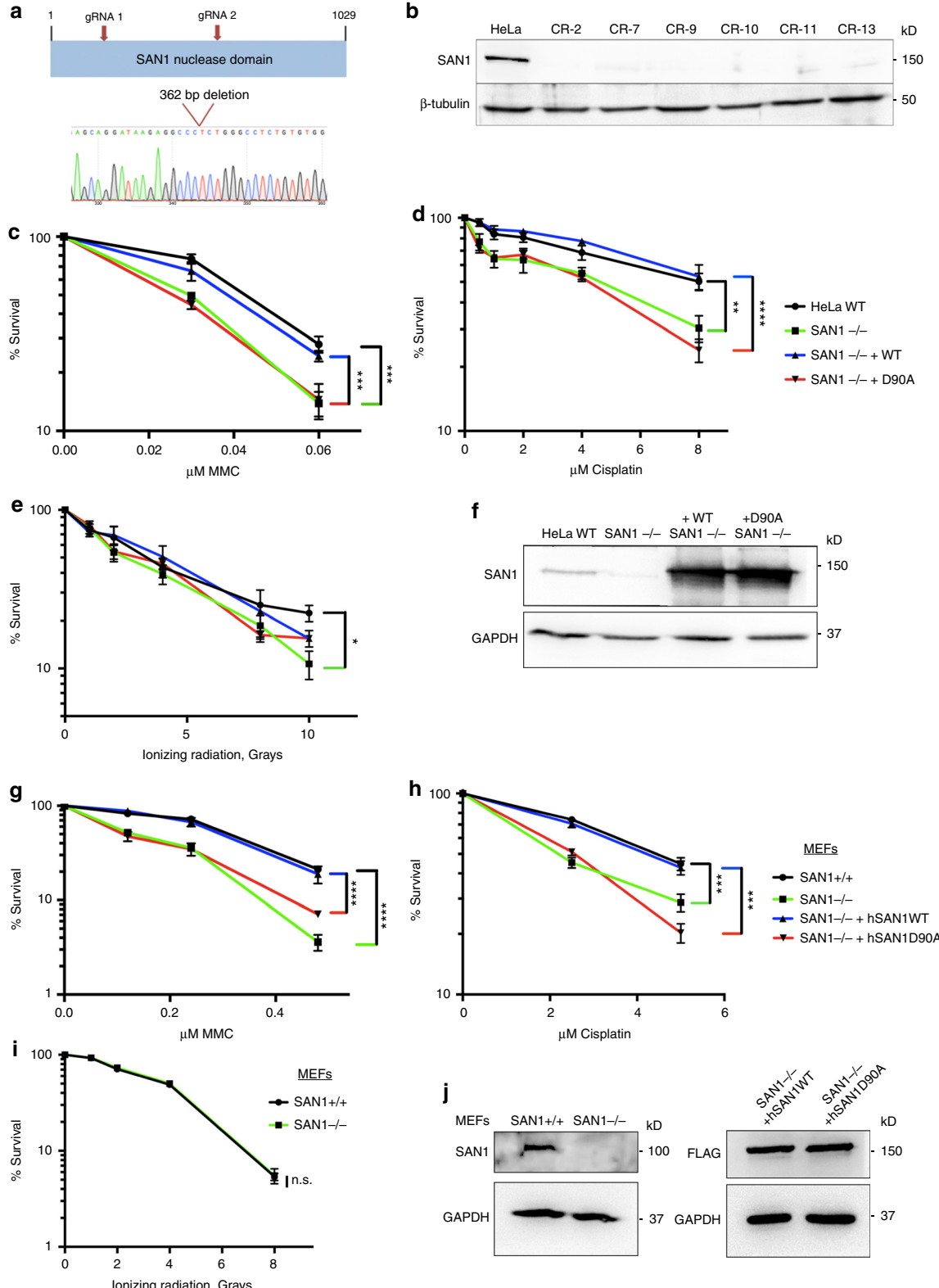

5′ end, and digests ssDNA. SAN1 also acts on splayed arms, but not other intermediates present in DNA repair such as HJs or nicked or gapped substrates. These biochemical properties are consistent with a model in which the enzyme binds DNA substrates downstream of where it cleaves, perhaps interacting with the other strand of a splayed arm substrate, and cuts 5′ to its binding site at a location of ~3 to ~7 nt from the 5′ end (Fig. 2f).

**SAN1 nuclease domain interacts with the C-terminal region**. The DisEMBL algorithm (http://dis.embl.de/) predicts the repeat motif region of SAN1 to be disordered, and therefore highly flexible, which we hypothesized might enable the N-termini and C-termini to interact. To test this prediction, we co-expressed a FLAG-tagged version of the human N-terminal nuclease domain and a Myc-tagged version of the mouse C-terminal domain, and asked if the two fragments could co-precipitate. Myc-tagged RhoA was used as a negative control. As shown in Fig. 2g, the FLAG-tagged Nuclease domain interacts robustly with the Myc-C-terminal domain (even though it is expressed at extremely low levels; see extended exposure in bottom left panel) but does not interact with Myc-RhoA (expressed at much higher levels).

To test whether the C-terminus or repeat regions are required for nuclease activity, we created deletion mutants (schematic in Figure 2h). ΔCterm-FLAG mutant showed no activity on a ssDNA substrate, while SAN1ΔRep-FLAG, which lacks the central repeats region, exhibited nuclease activity similar to that of the WT protein (Fig. 2h, i).

**SAN1 is required for efficient response to ICLs**. To probe the involvement of SAN1 in the DNA damage response we employed CRISPR/Cas9 editing of HeLa cells to disrupt the *Fam120b* gene. Two guide RNAs were used, to delete about half of the nuclease domain (Fig. 3a). Several clones were obtained that were homozygous deletions for the *Fam120b* gene, as determined by PCR and sequencing (Fig. 3a), and which did not express detectable SAN1 protein (Fig. 3b). The SAN1 KO cells proliferated at a similar rate to parental SAN1 WT cells (Supplementary Figure 3a) and were morphologically indistinguishable. Silencing of SAN1 expression by shRNA and siRNA also did not affect cell growth or cell cycle distribution (Supplementary Figure 3b, c). To determine whether loss of SAN1 alters sensitivity to DNA damage, we treated the KO and parental cells with various DNA damaging agents and performed colony survival assays (CSAs). There was a small reduction in number and size of SAN1 KO colonies as compared to the WT control, and notably, the KO cells were more sensitive to treatment with ICL agents MMC and Cisplatin (Fig. 3c, d). Loss of SAN1, however, only slightly increased sensitivity to ionizing radiation (IR) (Fig. 3e), which induces both ss breaks and DSBs. The effects on hydroxyurea and

camptothecin sensitivities, which cause replication stress, were also small to negligible (Supplementary Figure 3d, e), suggesting that SAN1 does not function in the repair of replication-stress induced DNA damage. Similar results were obtained using shRNAs against SAN1 in HeLa cells, where depletion of SAN1 only sensitized cells to cross-linking agents (Supplementary Figures 3f–i).

To test whether the nuclease activity of SAN1 is required for the response to ICLs, we created stable lines from the SAN1 KO cells that express WT ssf-tagged SAN1 or the nuclease-deficient D90A mutant. Both SAN1 WT and D90A display similar localizations in the cytoplasm and nucleus (Fig. 3f, Supplementary Figure 3j). Functional, WT SAN1-ssf fully rescued the survival defect of SAN1−/− cells treated with MMC or Cisplatin, but the D90A mutant failed to do so (Fig. 3c, d).

As an independent test for a role of SAN1 in the response to DNA damage, we created a conditional KO mouse, and generated mouse embryonic fibroblasts (MEFs). SAN1−/− MEFs showed significantly increased sensitivity to ICL agents as compared to WT cells (Fig. 3g, h), and were not sensitive to IR (Fig. 3i). Moreover, re-introduction of human SAN1-ssf WT rescued survival but the D90A mutant did not (Fig. 3g, h, j).

Taken together, these data conclusively demonstrate that SAN1 nuclease activity is required for survival in response to ICL agents, but not to those that induce DSBs, suggesting that it plays a role in ICL repair but does not participate in HR or non-homologous end joining (NHEJ).

**SAN1 deficiency phenocopies defects in the FA pathway**. A diagnostic feature of Fanconi anemia is the formation of radial chromosomes in hematopoietic cells following treatment with cross-linking agents[25]. These structures form when unrepaired ICLs cause replication fork collapse and one-sided DSBs, which are aberrantly joined to other chromosomes, leading to DNA breakage, chromosomal aberrations, and cell death (Deans and West[4]). No radials or chromosomal aberrations were found in either WT or SAN1−/− HeLa cells in the absence of treatment (Fig. 4a, b). Strikingly, however, metaphase spreads of the SAN1 −/− HeLa cells revealed a substantial increase in radial chromosome formation and other chromosomal aberrations as compared to parental cells after treatment with 30 nM MMC (Fig. 4c–f). These data, in conjunction with the specific sensitivity of SAN1−/− cells to ICL agents, strongly implicate SAN1 in the response to ICLs.

We also analyzed HeLa WT and SAN1−/− cells for DNA damage by examining levels of γH2AX. As shown in Fig. 5a, b, the amount of γH2AX was slightly higher in the SAN1−/− cells than the parental controls in untreated conditions, but at 48 and 72 h following MMC treatment, SAN1−/− cells displayed much higher levels of γH2AX (Fig. 5a, b). These results are consistent with the increased levels of radial chromosomes and aberrations

**Fig. 3** Loss of SAN1 leads to sensitization of cells to ICLs. **a** Schematic showing CRISPR-Cas9 strategy to create SAN1-/- HeLa cell lines. Two guide RNAs were used to delete a 362 bp region of exon 1 in the *fam120b* gene locus, which contains the conserved FEN1 family nuclease domain. **b** Immunoblot of HeLa WT parental cell line and CRISPR-Cas9 generated SAN1-/- cell lines showing loss of SAN1 expression (lanes 1–7). β-tubulin was used as a loading control. **c–e** SAN1-/- cells were transduced with lentiviral constructs expressing Strep$_2$-FLAG tagged SAN1 WT or the D90A mutant to create stable rescue cell lines. Colony survival assays (CSAs) were then performed using HeLa WT, SAN1−/−, and WT or D90A rescue lines with MMC, Cisplatin, or ionizing radiation ($N > 3$). Statistical significance determined by two-way ANOVA comparing HeLa WT and SAN1−/− or SAN1−/− +WT and SAN1-/- +D90A. Error bars denote s.e.m. $*p < 0.05$, $**p < 0.01$, $***p < 0.001$, $****p < 0.0001$. **f** Immunoblot showing SAN1 expression in HeLa WT, SAN1−/−, and SAN1 WT and D90A rescue lines. GAPDH was a loading control. **g–i** SAN1+/− mice were crossed to generate SAN1+/+ and −/− mouse embryonic fibroblasts (MEFs). The MEFs were immortalized using SV40 large T antigen, and the SAN1−/− MEFs were transduced with lentiviral constructs containing Strep$_2$-FLAG-tagged human SAN1 WT or SAN1 D90A. These cell lines were then used for CSAs with MMC, Cisplatin, or ionizing radiation ($N = 3$). **j** Immunoblot showing mouse SAN1 expression in SAN1+/+ and −/− MEFs (left panel) and hSAN1WT and hSAN1D90A expression in SAN1−/− cells (right panel). GAPDH was used as a loading control. Statistical significance for CSAs was determined by two-way ANOVA test comparing SAN1+/+ and SAN1−/− or SAN1−/− +hSAN1WT and SAN1−/− +hSAN1D90A

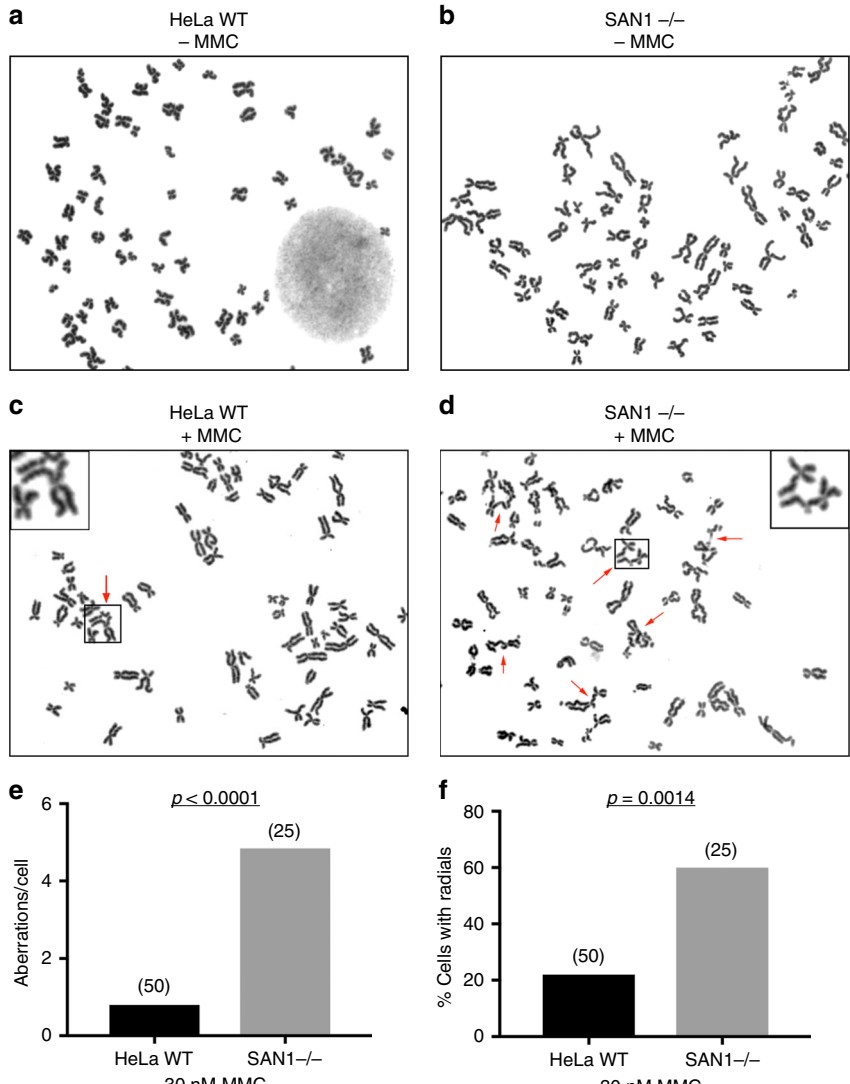

**Fig. 4** SAN−/− cells display increased levels of radial chromosomes in response to MMC. **a, b** Micrographs of metaphase spreads from untreated HeLa WT and SAN1−/− cells. **c, d** Micrographs of metaphase spreads of HeLa WT and SAN1−/− cells following treatment with 30 nM MMC, showing large increase in radials or other chromosomal aberrations in SAN1−/− cells. Red arrows indicate radial chromosomes or aberrations. **e, f** Quantification of aberrations/cell and percentage of radials/cell for HeLa WT and SAN1−/− cells treated with MMC. Metaphase spreads from 50 HeLa WT cells were analyzed (11 radial forms, 11 cells with radials, 40 aberrations). Metaphase spreads from 25 SAN1−/− cells were analyzed (42 radial forms, 15 cells with radials, 121 aberrations). Data were analyzed in Prism GraphPad from contingency tables using Fisher's exact-test (two-sided $p$ value)

(Fig. 4c–f) that result from unrepaired ICLs. In agreement with these data, we also detected a large and significant increase in γH2AX staining by immunofluorescence in SAN1−/− cells after 30 h with a higher concentration of MMC (Fig. 5c, d). Although we observed a smaller but significant increase in γH2AX intensity in untreated conditions, this might be a result of the presence of endogenous cross-links that form from sources including aldehydes[26].

Finally, the deletion of SAN1 increased 53BP1 foci (another marker of DSBs) following treatment with MMC (Fig. 5e, f)[27]. Collectively, these data demonstrate that SAN1 is required for a normal cellular response to ICLs, and its absence results in elevated DNA damage, DSBs, and chromosomal aberrations.

**SAN1 functions independently of the FA pathway**. We next asked if SAN1 functions within the FA pathway, the canonical mechanism for ICL repair, using epistasis experiments in which we first depleted a central component of the FA pathway,

FANCD2. FANCD2 plays a critical role in the activation of the FA pathway, and acts upstream of proteins that function in the nucleolytic steps of ICL repair such as SLX1-SLX4, XPF-ERCC1, and FAN1[9,28,29]. Treatment of WT HeLa cells with siRNAs against FANCD2 led to a sharp decrease in cell survival when cells were exposed to low doses of Cisplatin or MMC, as previously reported (Fig. 6a–c)[30]. Strikingly, when FANCD2 was depleted in SAN1−/− cells, a synergistic increase in the sensitivity of SAN1−/− cells to ICLs occurred. Additionally, we examined if SAN1 might be epistatic to XPF (FANCQ), the 3′ Flap endonuclease that forms a dimer with ERCC1 to unhook ICLs[9,10]. Similarly to FANCD2 depletion, we observed that XPF deficient cells were highly sensitive to ICLs, and that loss of XPF resulted in a greater sensitivity to ICLs in SAN1−/− cells than in HeLa WT cells (Supplementary Figures 4a–c).

In these experiments we observed only a small to negligible decrease in survival between HeLa WT and SAN1−/− cells at very low concentrations of Cisplatin and MMC. This result,

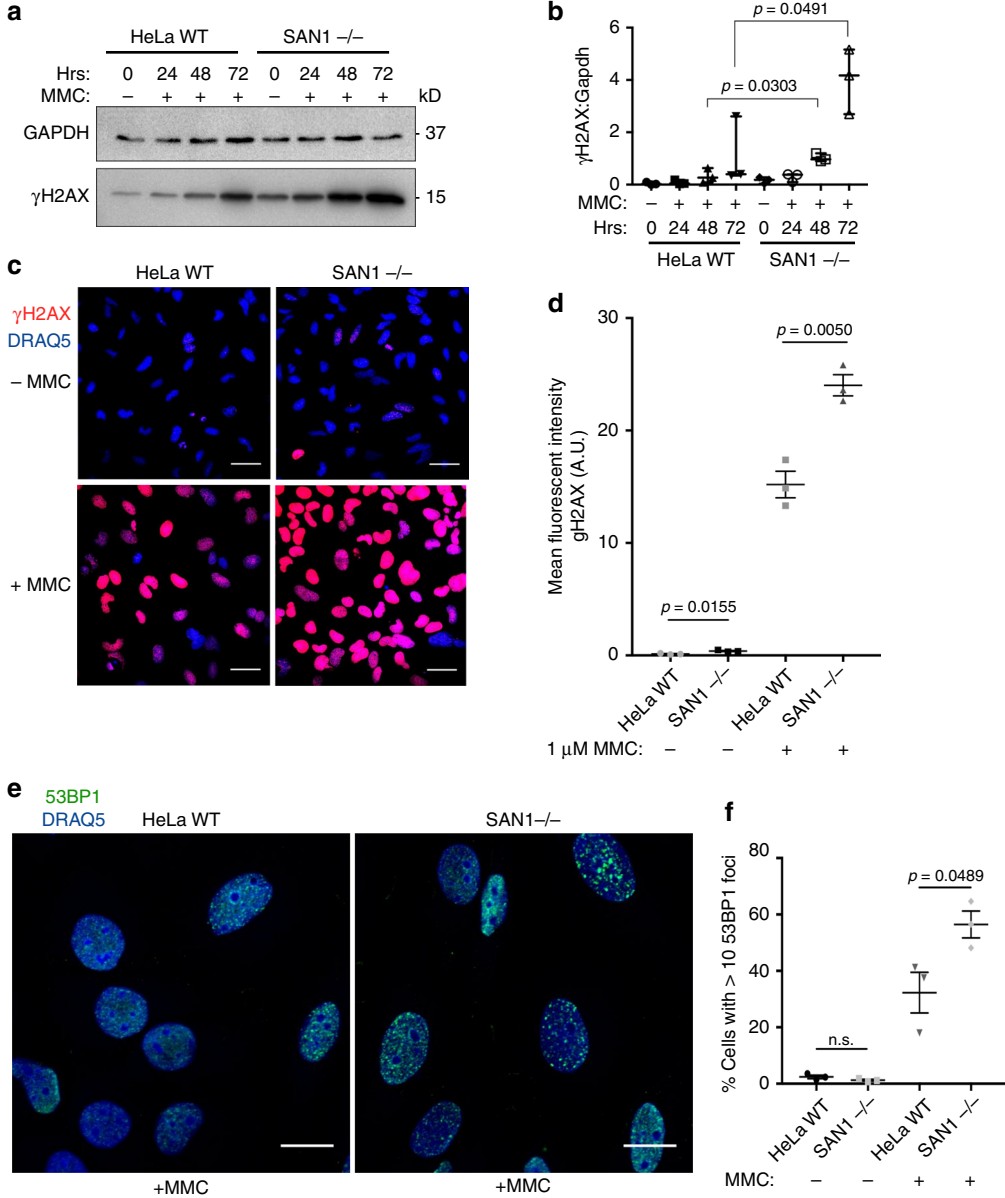

**Fig. 5** Increased levels of DNA damage in SAN1−/− cells exposed to MMC. **a** Immunoblots of γH2AX from HeLa WT and SAN1−/− cells treated for various times with vehicle or 0.045 μM MMC. GAPDH was a loading control. **b** Quantification of immunoblot time course (N = 3). Statistical significance determined by t-test with Welch's correction. **c** Immunofluorescence (IF) staining of γH2AX and DNA (Draq5) 30 hrs after treatment with vehicle or 1 μM MMC. **d** Quantification of γH2AX intensity in vehicle and MMC treated HeLa WT and SAN1−/− cells. Statistical significance determined by t-test with Welch's correction (N = 3 biological replicates, at least 250 cells per sample were analyzed). **e** IF staining of 53BP1 and DNA (Draq5) 30 h post-treatment with vehicle or 1 μM MMC. **f** Quantification of percentage of cells with >10 53BP1 foci in HeLa WT and SAN1−/− cells. Statistical significance determined by unpaired t-test (N = 3 biological replicates, at least 200 cells per sample were analyzed)

combined with the synergistic increase in sensitivity after depletion of FANCD2 and XPF in SAN1−/− cells, suggests that ICLs are predominantly processed by the FA pathway and that SAN1 might function in a secondary pathway if the FA pathway is overwhelmed by abundant ICLs. Collectively, our data argue that that SAN1 is not epistatic to FANCD2 or XPF, and functions independently of the FA pathway (Fig. 6a–c, Supplementary Figure 4a–c).

We also asked whether loss of SAN1 might indirectly decrease survival of MMC-treated cells, by affecting FA pathway activation. To explore this possibility, we examined FANCD2 mono-ubiquitylation and focus formation in response to MMC. However, loss of SAN1 had no impact on focus formation

(Fig. 6d, Supplementary Figure 4d–e), and mono-ubiquitylation following MMC treatment was normal (Fig. 6e), indicating that the FA pathway remains intact in SAN1−/− cells.

The FA pathway is a replication-dependent ICL repair mechanism activated when two replication forks collide with an ICL[3]. Because SAN1 acts independently of FA proteins, we asked if SAN1 might be redundant with other 5′ nucleases that function in ICL repair separately from the FA pathway. The SNM1A nuclease can participate in TCR of ICLs[31], or through an interaction with the DNA replication protein PCNA[32]. SNM1A does not interact with FANCD2 and has not been shown to function downstream of the FA pathway[33]. We depleted SNM1A in HeLa WT and SAN1−/− using siRNAs (Fig. 6h), and

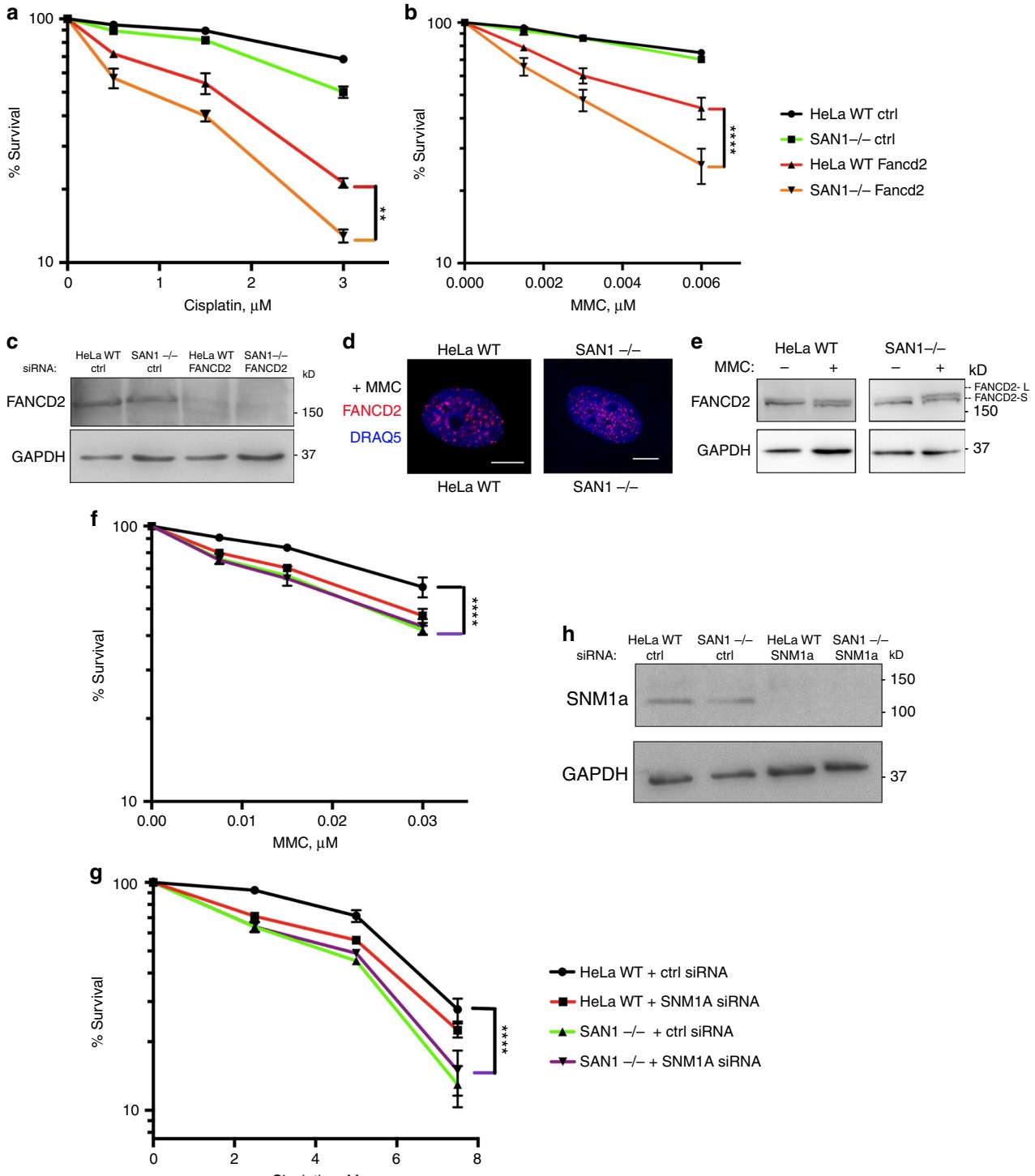

**Fig. 6** SAN1 functions independently of the FA pathway and does not affect FA pathway activation. **a**, **b** CSAs of HeLa WT and SAN1−/− cells treated with scrambled ctrl siRNA or FANCD2 siRNA, in response to Cisplatin and MMC ($N = 3$). Statistical significance determined by two-way ANOVA. Error bars denote s.e.m. *$p < 0.05$, **$p < 0.01$, ***$p < 0.001$, ****$p < 0.0001$. **c** Immunoblot showing siRNA knockdown of FANCD2 in HeLa WT and SAN1−/− cells. **d** IF staining of FANCD2 foci in HeLa WT cells and SAN1−/− cells treated with 0.045 μM MMC. **e** Immunoblot of FANCD2 showing mono-ubiquitylation in HeLa WT and SAN1−/− cells treated with vehicle or 0.045 μM MMC. **f**, **g** CSAs of HeLa WT and SAN1−/− cells treated with ctrl or SNM1A siRNA and exposed to Cisplatin or MMC. Statistical significance was determined by two-way ANOVA test. **h** Immunoblot of SNM1A in HeLa WT and SAN1−/− cells treated with ctrl or SNM1A siRNA

performed CSAs in response to MMC and Cisplatin (Fig. 6f, g). In HeLa WT cells depleted of SNM1A, we observed mild sensitivity to both both agents, as has been previously reported[34]. Importantly, we observed no further increase in the sensitivity of SAN1−/− cells depleted of SNM1A, indicating that SAN1

functions epistatically with SNM1A in preventing ICL sensitivity (Fig. 6f–h).

We also depleted the nuclease FAN1 by RNAi in both HeLa WT and SAN1−/− cells. FAN1, a 5′ nuclease involved in ICL repair, interacts with FANCD2 but this interaction is not

necessary for its role in processing ICLs[35]. Similarly to SNM1a depletion, depletion of FAN1 did not result in further sensitivity to MMC in SAN1−/− cells (Supplementary Figure 4f, g).

We conclude that SAN1 functions independently of the FA pathway in ICL repair, and that loss of SAN1 does not affect FA activation. Furthermore, SAN1 might participate in an alternative repair pathway with the 5′ nucleases SNM1a and FAN1, which also act independently of the FA pathway.

## SAN1 interacts with the helicase senataxin.

To probe the function of SAN1 we sought potential interacting proteins using a Hybrigenics genome-wide yeast two-hybrid (Y2H) screen. Full length murine SAN1 was used as bait, and was screened against a

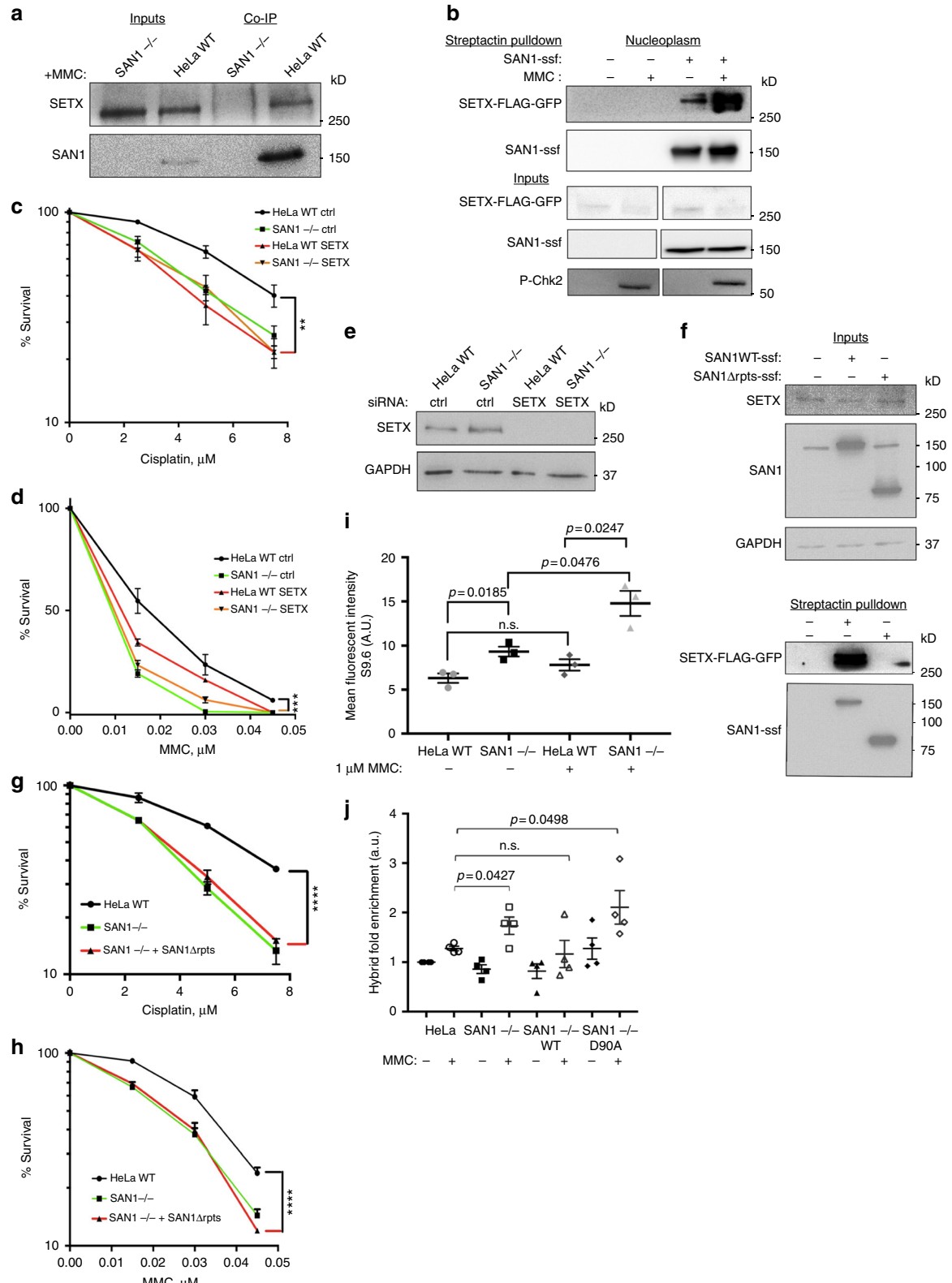

mouse adult brain library with a complexity of $\sim 1 \times 10^7$, at 10-fold coverage. From 183 positive clones, 9 were in-frame and were present more than once (Supplementary Figure 5a). Amongst these high-confidence hits was Senataxin (SETX), a gene mutated in ataxia oculomotor apraxia type 2 (AOA2) and juvenile amyotrophic lateral sclerosis (ALS4)[35,36]. SETX has been shown to participate in several different DNA damage responses, and functions centrally as a RNA/DNA helicase in the resolution of R-loops[37]. A small region of this protein, mSETX (876–1103), N-terminal to its helicase domain, bound to SAN1 (Supplementary Figure 5a). Based on the role of Senataxin in various areas of the DNA damage response, we explored whether an interaction between SAN1 and Senataxin is important for SAN1 function.

Immunoprecipitation of endogenous SAN1 co-precipitated a band detected by an anti-SETX antibody from HeLa WT cell nuclear extracts, but not from SAN1−/− cells (Fig. 7a). We also observed this interaction using a stable cell line expressing a near endogenous level of SETX-FLAG-GFP[38], and expressing SAN1-ssf (Fig. 7b). Importantly, treatment of the cells with MMC strongly increased the binding of SETX-FLAG-GFP. The promotion by MMC of the interaction between SAN1 and SETX suggests that they may function together in ICL repair.

Despite roles in various types of DNA damage responses[39–42], there is little evidence for any involvement of SETX in ICL repair. To test if SETX is involved in the response to ICLs, and whether SAN1 and SETX function together, we performed an epistasis experiment by depleting SETX with siRNA in either WT or SAN1−/− cells (Fig. 7c–e). Depletion of SETX in HeLa WT cells led to increased sensitivity to Cisplatin similar to that of SAN1−/− cells, and a mild sensitivity of cells treated with MMC (Fig. 7c–e). Notably, the depletion of SETX in SAN1−/− cells did not further increase sensitivity to either MMC or Cisplatin, suggesting that SAN1 might function cooperatively with SETX in response to ICLs.

To further investigate the importance of this interaction, we asked if binding to SETX is required for SAN1 function. We mapped the region of SAN1 required to bind SETX using a Myc-tagged SETX fragment that was shown to interact with SAN1 from our Y2H screen. A SAN1 mutant lacking the central repeat region (SAN1ΔRep-Flag) was unable to bind the Myc-mSETX fragment (Supplementary Figure 5e) although, importantly, it retained full nuclease activity (Fig. 2h, i). Moreover, unlike SAN1WT-ssf, SAN1ΔRep-ssf was unable to interact with full length SETX-Flag-GFP (Fig. 7f). We next used this mutant to test if SAN1 binding to SETX is essential for ICL resistance. Strikingly, the SAN1ΔRep-ssf mutant was unable to rescue survival in SAN1−/− cells treated with Cisplatin or MMC (Fig. 7g, h).

Finally, given the known role of Senataxin in resolving R-loops, we asked whether the loss of SAN1 might lead indirectly to R-loop accumulation, from a failure to repair and remove ICLs. To test this possibility, we performed immunofluorescence staining of HeLa WT and SAN1−/− cells using the S9.6 monoclonal antibody, which specifically detects RNA–DNA hybrids. Co-staining with nucleolin antibody was used to subtract the constitutive presence of RNA-DNA hybrids in the nucleoli. No significant difference in R-loops was found between untreated and MMC treated HeLa WT cells. However, the abundance of R-loops in SAN1−/− cells after MMC treatment was significantly increased (Fig. 7i, Supplementary Figure 5b). We also observed a small increase in the levels of R-loops in untreated SAN1−/− cells compared to HeLa WT cells, which might be attributable to the presence of ICLs from endogenous sources such as aldehydes[26], or to the possible regulation of SETX activity by SAN1 binding.

To validate the specificity of R-loop staining, fixed and permeabilized HeLa WT and SAN−/− cells were treated with ribonuclease (RNase) H, which completely abolished the S9.6 signal (Supplementary Figure 5c). As a positive control we showed that, as previously reported[37], silencing of the SETX helicase was sufficient to increase R-loop abundance (Supplementary Figure 5d).

To provide an independent assessment of R-loop formation, we used a dot blot assay in which isolated genomic DNA is probed for RNA/DNA hybrids with the S9.6 antibody, and normalized to total ssDNA (Fig. 7j). Treatment of cells with MMC caused a significant increase in R-loops in SAN1−/− cells compared to WT cells. Moreover, the re-expression of WT SAN1, but not of the nuclease-defective D90A mutant, blocked any increase in R-loop formation in response to MMC (Fig. 7j). We did not observe a similar increase between SAN1−/− cells compared to HeLa WT cells in untreated conditions as in the S9.6 staining in Fig. 7i; however this might reflect differences in assay sensitivity or dynamic range.

To further probe the relationship between SAN1, R-loops, and ICLs, we over-expressed RNaseH1-NLS-mCherry to degrade R-loops in the nucleus, and examined the response to MMC in HeLa WT and SAN1−/− cells. Over-expression of RNaseH1 caused an increase in γH2AX intensity in response to MMC in HeLa WT cells, while γH2AX in SAN1−/− cells was reduced to levels comparable to the WT cells (Supplementary Figure 6c, d). We speculate that the formation of R-loops adjacent to ICLs in HeLa WT cells might be necessary for SAN1 to participate in a repair process required to prevent or repair DNA damage, possibly through the recruitment of SAN1 through its interaction with Senataxin (Supplementary Figure 7). Additionally, the

**Fig. 7** SAN1 interacts with the RNA/DNA helicase sSenataxin. **a** Endogenous SAN1 was co-immunoprecipitated from HeLa WT and SAN1−/− after treatment with 1 μM MMC. Top panel: immunoblot (IB) of Senataxin inputs (lanes 1–2) and Co-IP (lanes 3–4), bottom panel: IB of SAN1 inputs (2%) (lanes 1–2), and Co-IP (lanes 3–4). **b** A stable HeLa cell line expressing near endogenous levels of a Senataxin-FLAG-GFP construct was transduced with a lentiviral construct of SAN1WT-Strep$_2$-FLAG (SAN1ssf). Soluble nuclear fraction was isolated from the cells and SAN1 was captured on Strep-Tactin beads. Top panel: IB for Senataxin and SAN1 of precipitations from HeLa SETX-FLAG-GFP cell line +/- SAN1-ssf and +/- MMC. Bottom panel: Input IB for Senataxin, SAN1 and P-Chk2 from HeLa SETX-FLAG-GFP cell lines +/- SAN1-ssf, and +/- 1μM MMC. **c, d** CSAs of HeLa WT and SAN1−/− cells, transfected with scrambled ctrl or SETX siRNAs, in response to Cisplatin and MMC. Statistical significance determined by two-way ANOVA. Error bars denote s.e.m. *$p < 0.05$, **$p < 0.01$, ***$p < 0.001$, ****$p < 0.0001$. MMC CSA is shown in linear scale in **d** due to zero values at higher MMC concentrations. **e** IB of SETX siRNA knockdown. **f** Cells were fractionated to prepare the soluble nuclear fraction as in **b** and was captured on Strep-Tactin beads. Upper panel: IB of inputs for stable HeLa cell lines expressing near endogenous levels of a Senataxin-FLAG-GFP construct and over-expressing SAN1WT-Strep$_2$-FLAG (SAN1ssf) or SAN1 lacking the central repeats region (SAN1ΔRep-ssf). Lower panel: co-immunoprecipitation of SETX-FLAG with SAN1WT-ssf but not SAN1ΔRep-ssf. **g, h** CSAs for HeLa WT, SAN1−/−, and SAN1−/− +SAN1ΔRep-ssf cells exposed to Cisplatin and MMC. Statistical significance determined by two-way ANOVA. **i** Quantification of nuclear R-loop intensity ($N = 3$). HeLa WT and SAN1−/− cells were treated with vehicle or 1 μM MMC and labeled with a monoclonal antibody to detect RNA/DNA hybrids (S9.6), nucleolin, and Draq5. Statistical significance calculated using unpaired t-test ($N = 3$ biological replicates, at least 60 cells per sample were analyzed). **j** Dot blot assay for quantification of RNA/DNA hybrids. ($N = 4$) Statistical significance determined by unpaired t-test comparing each condition to HeLa WT untreated

reduction of γH2AX in MMC treated SAN1−/− cells over-expressing RNaseH1, might result from of the removal R-loops that contribute to DNA damage as a consequence of unrepaired ICLs.

Given the involvement of Senataxin in TCR processes such as R-loop resolution, we investigated if the increased levels of DNA damage present in SAN1−/− following MMC treatment might be dependent upon transcription. To test this hypothesis we treated HeLa WT and SAN1−/− cells with MMC and the RNA pol II inhibitor α-amanitin for 6 hrs, and examined the percentage of γH2AX positive cells (Supplementary Figure 6a, b). There was a small increase in the percentage of γH2AX positive cells in SAN1−/− cells compared to HeLa WT, but substantially less than that observed when the cells were treated for a longer time period with MMC (Fig. 5a–d). However, α-amanitin decreased the percentage of γH2AX positive cells for both HeLa WT and SAN1−/− cells in the presence of MMC. To further explore the possibility that SAN1 might participate in a more general TCR process we also examined if SAN1−/− cells are sensitive to UV radiation, but we detected no increased sensitivity compared to HeLa WT cells (Supplementary Figure 6c). Based on these results we are unable to conclude if SAN1 functions specifically in TCR or some other replication-independent repair process in response to ICLs.

We conclude that SAN1 binding to the RNA/DNA helicase SETX is required for SAN1 to function in the response to ICLs. SAN1 and SETX likely cooperate in the same pathway, as depletion of SETX is epistatic with loss of SAN1. Consistent with this model, loss of SAN1 leads to an increase in R-loops following treatment with MMC, likely as a result of the inability of SAN1 to participate in a step of ICL repair.

## Discussion

We have discovered that Fam120b, previously described as a transcriptional co-activator of the peroxisome proliferator-activated receptor gamma[20], contains an N-terminal domain closely related to the FEN1 family of nucleases, and is in fact an exonuclease that cleaves the 5′ end of ssDNA and splayed arm structures. Fam120b has never been identified in proteomics or two-hybrid screens for factors associated with the FA pathway or other DNA repair pathways. Therefore, we sought other factors with which it might be involved. A Y2H screen revealed an interaction with the RNA/DNA helicase SETX. Therefore, we named this enzyme SAN1 (Senataxin-associated nuclease 1). Strikingly, the SAN1-SETX interaction is essential for SAN1 function in preventing cross-link sensitivity. SETX plays an important role in TCR through the resolution of R-loops, and cells lacking SAN1 exhibit higher levels of R-loops, most likely as a result of the inability to remove ICLs from sites where transcription complexes have stalled.

Several lines of evidence support our conclusion that SAN1 functions in ICL repair. (1) Cells lacking SAN1 show decreased survival in response to ICLs; (2) such cells also incur elevated DNA damage as detected by increased γH2AX levels and p53BP1 foci; (3) SAN1−/− cells display elevated levels of radial chromosomes and chromosomal aberrations in response to MMC, a diagnostic feature of Fanconi anemia; and (4) SAN1 is epistatic to the SNM1A and FAN1 nucleases, which participate in ICL repair[31]. We conclude that SAN1 is required for an efficient cellular response to ICLs. Unlike the nucleases FEN1 and EXO1, SAN1 is not essential for the repair of DSBs induced by IR, arguing that SAN1 is not required for HR or NHEJ[31,43,44]. Moreover, unlike EXO1, which cleaves single nucleotides from the 5′ ends of its substrates, SAN1 removes ~3 or ~7 nt units. It can use either ssDNA or splayed forks as substrates, and requires a free 5′ end.

Our data also suggest that SAN1 does not act in the FA pathway, as SAN1 is not epistatic to XPF or FANCD2. SAN1 deletion also had little effect on sensitivity to agents that increase replication stress, such as hydroxyurea and camptothecin. Rather, SAN1 associates with SETX, an RNA/DNA helicase. This association is increased in the presence of ICLs; and R-loop formation in the presence of ICL agents is promoted by loss of SAN1, which we suggest results from increased stalling of transcriptional complexes at unrepaired ICLs.

It is unlikely that SAN1 plays any direct role in R-loop resolution adjacent to ICLs, given that no evidence exists that persistent R-loops can lead to radial chromosomes that we detect in MMC-treated SAN1−/− cells. Instead these structures are specific to a defect in ICL repair, and result from collapsed replication forks and one-sided DSBs[4]. The inability to properly respond to ICLs in SAN1−/− cells might lead to greater transcriptional stalling at ICLs, thereby generating R-loops, as well as additional stalled replication forks at ICLs that the FA pathway cannot process. We speculate that the interaction of SAN1 with SETX might be necessary for SAN1 recruitment to sites where it can participate in a repair process required for the removal of ICLs (Supplementary Figure S7).

Collectively these studies support the existence of a repair pathway in which SAN1 acts independently of the FA pathway to protect cells from ICLs. SAN1 might act in a backup pathway that is required when the FA pathway is overwhelmed. It is tempting to speculate that SAN1 acts with SETX at ICL sites where transcription complex stalling has occurred, and the subsequent formation of R-loops recruits SETX, which in turn recruits SAN1. The formation of a ssDNA break near an RNA:DNA hybrid could lead to a 5′ splayed arm-like substrate after resolution of the R-loop by SETX. This would allow for digestion of the ssDNA flap adjacent to an ICL by SAN1, possibly providing a better substrate for the 5′ exonuclease SNM1A, which might unhook the lesion by digesting past the cross-link (Supplementary Figure S7).

Further studies will be important to investigate the regulation of SAN1, as well as the specific step within the ICL repair process at which SAN1 acts.

## Methods

**Plasmids and siRNA.** Human SAN1 cDNA (FLJ56631) was purchased from the NBRC, Japan, and cloned into the pRK7 expression vector with a C-terminal FLAG tag. SAN1 (D90A) and SAN1 rescue constructs were made by QuikChange site-directed mutagenesis (Agilent Technologies). Partial mouse SAN1 open reading frames were obtained from GE Healthcare Dharmacon MGC cDNAs library and cloned into pASK-IBA3Plus to express full length SAN1-Strep-tag II. The D90A mutation was introduced using Stratagene QuikChange. pICE-RNaseHI-WT-NLS-mCherry was a gift from Patrick Calsou (Addgene plasmid # 60365). For RNAi knockdown experiments SMARTpool siGENOME (Thermo Fisher and Dharmacon) siRNAs were used for FANCD2 (M-016376-02-005), SETX (M-021420-01-0005), SNM1A (L-010790-00-0005), XPF (M-019946-00-0005), FAN1 (L-020327-00-0005), and FAM120B (M-014898-00-0005). Cells were transfected, split 24 h later, and transfected again, with specific siRNA pools or a scrambled control (D-001810-01-05) using RNAiMAX (Invitrogen) and Opti-MEM media (Thermo Fisher) according to manufacturer's instructions. Colony survival assays were performed and lysates made 36 h after the second round of siRNA transfection. Knockdown was determined by western blotting using FANCD2 (1:500, Novus Biologicals NB100-182), SETX (1:1000, Novus Biologicals NB100-57542), SNM1a (1:1000, Bethyl Labs A303-747A-M), XPF (1:1000, Bethyl Laboratories A301-315A), FAN1 (1:1000, non-commercial), FAM120b (1:1000, Abcam ab106455).

**Cell culture and transfections.** HeLa (ATCC) and 293T (ATCC) cells were grown in DMEM supplemented with 10% FBS, 100 U/mL penicillin and 100 U/mL streptomycin (GIBCO). Transfection of plasmids or siRNA was performed with calcium phosphate, Lipofectamine 2000 (Invitrogen), or Lipofectamine RNAiMAX (Invitrogen). Virus was made, collected and titered as described previously[45].

**Colony survival assays.** HeLa or mouse embryonic fibroblast cells were seeded at 300–400 cells per six-well dish for ~16 h overnight, treated with DNA damaging agents, and allowed to form colonies for 7–10 days. Colonies were fixed in ice cold 70% EtOH, stained with crystal violet, and counted.

**Recombinant murine SAN1 expression and purification.** The pASK-SAN1 plasmids were transformed into BL21-CodonPlus (DE3)-RIPL bacterial cells. SAN1 expression was induced by the addition of tetracycline (0.2 μg/mL) to 0.5 L of bacteria in early exponential phase in liquid culture followed by overnight incubation at 18 °C with shaking at 250 rpm. Cells were centrifuged (3795x$g$, 20 min, 4 °C) and the pellet was resuspended in lysis buffer (50 mM Tris HCl pH 7.4, 150 mM NaCl, 1 mM MgCl$_2$, 1 mM PMSF, protease inhibitors (Roche)). Lysates were treated with lysozyme (1 mM, 4 °C, 15 min) before being ruptured in a French Press (1500 psi) twice. Lysates were then centrifuged (16,100x$g$, 20 min, 4 °C). The supernatant was loaded onto a pre-equilibrated 1 mL CV Gravity flow Strep-Tactin Sepharose column (IBA). The column was washed with 20 CV buffer w (50 mM Tris HCl pH 7.4, 150 mM NaCl) and eluted with buffer e (50 mM Tris HCl pH 7.4, 150 mM NaCl, 1 mM MgCl$_2$, 2.5 mM desthiobiotin) in 2.5 mL fractions. Samples from each elution fraction were subjected to SDS-PAGE. Fractions containing SAN1 were combined and protein concentration was estimated by SDS-PAGE followed by brilliant blue staining in parallel with known amounts of BSA.

**Co-IP of N-terminus and C-terminus of SAN1.** Fifteen microgram of pKhSAN1-Nuclease-FLAG and 15 μg of pKMyc-mSAN1-Cterminus were co-transfected into 293T cells by calcium phosphate transfection. In samples where only pKhSAN1-Nuclease-FLAG or pKMyc-mSAN1-Cterminus was transfected, 30 μg of DNA was used. Cells were lysed in buffer containing 25 mM HEPES pH 7.4, 150 mM NaCl, 0.5% Triton-X 100, 0.5 mM EDTA, 1 mM MgCl$_2$, 2 mM DTT, 1 mM PMSF, 10 μg/mL leupeptin, 20 μg/mL aprotinin 24 h later. Cell lysates were then incubated with anti-FLAG M2 agarose (Sigma A2220-1ML) for 1 h at 4 °C rotating before three washes. Samples were boiled for 8 min in 4× LSB and analyzed by immunoblot. pKMycRhoA was co-expressed with pKhSAN1-Nuclease-FLAG as a negative control.

**Production of hSAN1 from mammalian cells.** pKSAN1WT-FLAG or pKSAN1D90A-FLAG constructs were transfected into 293T cells. At 24 h post-transfection cells were washed once with 1× PBS followed by addition of Lysis Buffer (25 mM HEPES pH 7.4, 10% glycerol, 1 mM EDTA, 1 mM DTT, 20 nM calyculin A (Sigma) and protease inhibitors (Roche)) and incubation on ice for 5 min. Samples were snap frozen in liquid nitrogen and subsequently thawed at 37 °C three times. NaCl was added to 300 mM. Lysates were then centrifuged at 16,100x$g$ for 20 min at 4 °C and the supernatant was added to an equal amount of Lysis Buffer plus 0.2% NP-40 and then centrifuged at 16,000x$g$ for 5 min at 4 °C. Lysates were added to washed mouse anti-FLAG M2 agarose (Sigma) for 1 h at 4 °C. Samples were washed 2× with Wash Buffer (Lysis Buffer plus 0.1% NP-40 and 150 mM NaCl) for 15 min per wash at 4 °C. Samples were then washed 2× with Elution Buffer (62.5 mM HEPES pH 7.4, 62.5 mM KCl, 5% glycerol, 1 mM DTT, 50 μg/mL BSA) for 15 min per wash at 4 °C, and SAN1 protein was eluted from beads with Elution Buffer + 150 ng/μL FLAG peptide (20 μL/10 cm, 60 μL/15 cm) by shaking (600 rpm) at 4 °C for 30 min. Protein concentration was estimated by SDS-PAGE followed by Coomassie brilliant blue staining in parallel with known amounts of BSA.

**$^{32}$P Labeling and oligonucleotide annealing.** Defined oligonucleotide structures (ssDNA, dsDNA, nicked and gapped structures, replication fork, splayed arm structure, 5′ and 3′ flap structures) were 5′-labeled with 5 pmol γ-[$^{32}$P] ATP using T4 polynucleotide kinase (NEB) or 3′-labeled with 5 pmol α-[$^{32}$P] CordycepinTP using DNA terminal transferase (20 U) in 1× TdT buffer, supplemented with 2.5 mM CoCl$_2$ in a 10 μL reaction at 37 °C, according to standard methods (NEB). Oligonucleotides were annealed in 6× annealing buffer (0.9 M NaCl, 90 mM sodium citrate) by slow cooling from 95 °C and purified from 12% native PAGE gel by the crush and soak method. Oligonucleotide sequences are given in Table 1.

**Nuclease assay standard reaction conditions.** All reactions were carried out in nuclease buffer (62.5 mM HEPES pH 7.4, 62.5 mM KCl, 5 % Glycerol, 1 μM DTT, 50 μg/mL BSA). Five micromolar of substrate was added to 4 μL of protein (27.5 nmol) or water and after the addition of Start Buffer (6 mM MgCl$_2$, 2 mM β-ME, 0.05 μg/μL BSA) reactions were incubated at 37 °C for 2 h (unless otherwise noted). Reactions were stopped using 2.5 μL of 5× Stop Buffer (15 mM EDTA, formamide, 0.005% Bromophenol blue, 0.005% Xylene cyanol, 5% glycerol) followed by boiling the samples for 3 min. Unless otherwise stated DNA substrate sequences for individual experiments are shown schematically in each figure and listed in Table 1. Reactions were analyzed on 12% denaturing PAGE gel (1× TBE, 7 M urea, 12% 19:1 acrylamide/bis-acrylamide 19:1).

**Kinetic characterization of hSAN1.** Reactions were carried out with varied concentrations of substrate (10–1000 nM) and 240 pmol enzyme in Elution Buffer. Reactions containing SAN1 WT and substrate were pre-warmed to 37 °C and initiated by the addition of Start Buffer. Reactions were sampled at three time points by addition of Stop Buffer, and processed using the Nucleotide Removal kit (Qiagen). Products were quantified using Ecoscint Original scintillation liquid (National Diagnostics) and a scintillation counter and were analyzed using GraphPad Prism.

**Double-affinity Strep-FLAG WT and D90A SAN1 purification.** HeLa SAN1−/− cells + SAN1 WT-SSF or SAN1 D90A-SSF cells were washed once with 1× PBS followed by addition of Lysis Buffer 1 (50 mM Tris HCl pH 7.4, 1 mM MgCl$_2$, protease inhibitors (Roche). Samples were snap frozen in liquid nitrogen and subsequently thawed at 37 °C three times. NaCl was added to 150 mM. Lysates were then centrifuged at 16,100x$g$ for 20 min at 4 °C and the supernatant was added to washed Strep-Tactin beads (IBA) and rotated for 1 h at 4 °C. Samples were washed 2× with Wash Buffer (50 μM Tris HCl pH 7.4, 150 mM NaCl, protease inhibitors) for 15 min rotating per wash at 4 °C. SAN1 protein was eluted from beads with Elution Buffer (50 mM Tris HCL pH 7.4, 150 mM NaCl, 1 mM MgCl$_2$, 2.5 mM desthiobiotin) by shaking (600 rpm) at 4 °C for 30 min. Samples were then added to equal volume of 2× Lysis Buffer 2 (50 mM HEPES pH 7.4, 20% glycerol, 2 mM EDTA, 2 mM DTT, 40 nM calyculin A (Sigma) and protease inhibitors (Roche) and added to washed mouse anti-FLAG M2 agarose (1:1000, Sigma F1804) rotating for 1 h at 4 °C. Samples were washed 1× with Wash Buffer (Lysis Buffer 2 plus 0.1% NP-40 and 150 mM NaCl) for 15 min rotating at 4 °C. Samples were then washed 1× with Elution Buffer (62.5 mM HEPES pH 7.4, 62.5 mM KCl, 5% glycerol, 1 mM DTT) for 15 min rotating at 4 °C. SAN1 protein was eluted from beads with Elution Buffer + 150 ng/μL FLAG peptide by shaking (600 rpm) at 4 °C for 30 min. Samples from each step of the double-affinity Strep-FLAG purification were visualized by SDS-PAGE followed by silver stain or immunoblot (using mouse M2 anti-FLAG antibody 1:1000, Sigma F1804) or were tested for nuclease activity using the nuclease filter-spin assay described previously.

**Electrophoretic mobility shift assay with SAN1 D90A.** $^{32}$P-labeled X4 (1 nM) was incubated with different concentrations of SAN1 D90A (0, 1, 5, 10, and 20 nM) or RPA (10 nM) in binding buffer (20 mM HEPES pH 7.4, 0.1% NP-40, 0.1 M KCl, 5 mM MgCl$_2$, 1% glycerol, 0.25 μg/μL BSA, 1 mM DTT) for 30 min at room temperature. Four microliter of 6× Ficoll loading dye was added to the reactions and 10 μL of each reaction was separated by electrophoresis on a 5% gel (37.5:1 polyacrylamide, 0.5 × TBE) at 30 V for 3 h in a cold room with pre-chilled 0.5× TBE. Gels were dried and visualized by autoradiography.

**Proliferation assay.** WT HeLa cells and SAN1−/− HeLa cells were plated in six-well plates at 100 cells/well and allowed to adhere. Each day a number of the wells were counted. Relative cell growth was determined by dividing each time point by day 1.

**Production of SAN1−/− HeLa cells.** Two 60-bp guide sequences (sgRNA1 F: 5′-TTTCTTGGCTTTATATATCTTGTGGAAAGGACGAAACACCGCAGGATA AGAGAGATGAAT-3′ and sgRNA2 F: 5′-TTTCTTGGCTTTATATATCTTGTG GAAAGGACGAAACACCGAGAAGCTCTGTGAGAGTCT-3′) were designed to target sites in the first exon of SAN1, to remove the majority of the nuclease domain. The sgRNA1 and sgRNA2 were cloned into gRNA cloning vectors and verified by sequencing. gRNA1, gRNA2, and NLS-hCas9-NLS constructs were transfected into HeLa cells using Lipofectamine 2000 (Invitrogen 11668-030) per Invitrogen protocol publication number MAN0007824 Rev 1.0 and seeded at single cell density on 15 cm dishes. Individual colonies were isolated by 0.25% trypsin and were plated separately in six-well plates to grow. Genomic DNA was isolated from WT HeLa cells and from 12 clones. Exon 1 of SAN1 was PCR amplified using SAN1 genomic PCR primers (SAN1 genomic F: 5′-ACTGATTAATTTATCTTT CTTTCCAGATCC-3′ and SAN1 genomic R: 5′-TCTGGGATTATGTCGTTGCC AAGGAGG-3′) and clones that showed a deletion were sequenced and analyzed by immunoblot (SAN1 1:1000). 6 SAN1−/− HeLa clones were identified and experiments in this study were completed with SAN1−/− HeLa Clone 2 unless otherwise specified. Uncropped western blots of knockout cell lines and rescue lines are shown in Supplemental Figure 8.

**Generation of mouse embryonic fibroblasts (MEFs).** A conditional KO mouse was created through the Texas A&M Institute for Genomic Medicine, using EUCOMM ES cells targeting the *fam120b* gene (ES cell clone HEPD0652_5_G10). A *fam120b*/ + male was crossed to a FLPer/ + female to delete the lacz/neo markers, the FLPer transgene was then removed by crossing to a +/+mouse, and the resulting floxed allele mice were crossed to produce homozygotes. These were then crossed with a Sox2-Cre mouse to obtain a global knockout of the allele. *Fam120b* +/+ and −/− MEFs were isolated from day 13.5 embryos, from matings of heterozygous parents. Embryos were incubated in 0.25% Trypsin overnight at 4 °C for digestion to single cells. Cells were plated and cultured in 10% FBS, 100 U/mL penicillin and 100 U/mL streptomycin (GIBCO). MEFs were immortalized by transfection with a plasmid encoding SV40 Large T antigen (Addgene 21826), followed by a 1/10 split of the cells 5–6 times. Genotypes were validated by genomic PCR and western blotting using a non-commercial rabbit polyclonal antibody against murine SAN1 (1:1000). Uncropped western blots of knockout cell lines and rescue lines are shown in Supplemental Figure 8.

All animal studies were performed in compliance with ethical regulations in Vanderbilt University and were approved by the Institutional Animal Care and Use Committee, Vanderbilt University

**Immunofluorescence staining and microscopy**. FANCD2 staining: Cells were seeded in 8 well chamber Labtek II slides and treated with vehicle or 120 ng/mL MMC for 24 h. Cells were washed twice with 1× PBS, and fixed in 4% paraformaldehyde for 15 min. Cells were then blocked in 3% BSA with 0.3% Triton TX-100 for 1 h, then incubated with rabbit anti-FANCD2 (1:500, Novus Biologicals NB100-182) antibody for 1 h at room temp. Cells were washed 3× for 5 min with 1× PBS then incubated with 594 Goat anti rabbit (1:500, Alexa Fluor R37117) secondary antibody, and DRAQ5 nuclear stain (1:2000, cell signaling tech. 4084 S) for 1 h at room temp. Slides were mounted in Fluoromount G and sealed with nail varnish. Confocal images were acquired using a Nikon A1R confocal microscope using a 60× oil lens (na 1.3). For R-loop (S9.6) and Nucleolin staining, HeLa WT and SAN1−/− cells were treated with vehicle or 1 μM MMC for 30 h, and fixed in 4% paraformaldehyde, blocked in 3% BSA for 1 h at room temp, and incubated with mouse monoclonal S9.6 (1:200, Kerafast ENH001) and rabbit polyclonal anti Nucleolin (1:1000, Abcam ab22758) antibodies overnight at 4 °C. Cells were washed 3× for 5 min with 1× PBS and then incubated with 488 Goat anti mouse (1:500, Alexa Fluor A-11029) and 594 Goat anti rabbit (1:500, Alexa Fluor R37117) secondary antibodies, and DRAQ5 nuclear stain (1:2000, Cell Signaling Tech. 4084 S) for 1 h at room temp. Intensity of nuclear R-loop staining was determined by masking with the DRAQ5 channel to select the nuclei, and subtraction of S9.6 staining from nucleolin-positive regions. Slides were then mounted in Fluoromont G and sealed. For γH2AX and 53BP1 staining cells were washed twice with 1× PBS, fixed in 4% paraformaldehyde for 15 min, blocked in 3% BSA with 0.3% Triton TX-100 for 1 h at room temp, and incubated with γH2AX (1:500, Ser139 EMD Millipore 05-636 clone JBW301) or 53BP1 (1:500, Bethyl Laboratories A300-272A) for 1 h at room temperature. Cells were washed 3× for 5 min with 1× PBS and then incubated with 488 Goat anti mouse (1:500, Alexa Fluor A-11029) or 594 Goat anti rabbit (1:500, Alexa Fluor R37117) secondary antibodies, and DRAQ5 nuclear stain (1:2000, Cell Signaling Tech. 4084 S) For the RNaseH1-mCHerry experiment, cells were transfected with the pICE-RNaseHI-WT-NLS-mCherry construct using lipofectamine 2000, and incubated with 10 μg/mL of doxycycline for 24 h to induce expression. Cells were then treated with vehicle 1 μM MMC for 30 h and stained for γH2AX and DRAQ5 as described above.

**Radial chromosome assays**. HeLa cells were incubated for 48 h in the presence or absence of MMC. Colcemid (0.1 g/mL) was added to the medium 2 h before the cells were collected. For each sample, 50 metaphases were analyzed for chromosomal abnormalities, as previously described[46]. Data were analyzed in Prism GraphPad from contingency tables using Fisher's exact-test (two-sided p-value)

**Co-immunoprecipitation of SAN1 and Senataxin**. HeLa cells were fractionated as previously described (Huang et al. 2009), with the following modifications. Cells were washed once with 1× PBS and the pellet was resuspended in 2–5 volumes of lysis buffer (10 mM HEPES pH 7.4, 10 mM KCl, 0.05% NP-40) containing Roche protease and phosphatase inhibitors, and incubated on ice for 20 min. Lysates were centrifuged at 16,100×g at 4 °C for 10 min. The supernatant containing the cytoplasmic proteins was discarded, and the pellet containing the nuclei was washed once with lysis buffer and resuspended in Low Salt Buffer (10 mM Tris-HCl pH 7.4, 0.2 mM MgCl$_2$), incubated on ice for 15 min, and centrifuged for 10 min at 4 °C. The supernatant was removed and combined with an equal volume of 2× Co-IP buffer (20 mM HEPES pH 7.4, 200 mM NaCL, 1% Triton Tx-100, 1 mM EDTA, 10 mM MgCl$_2$) and placed on ice. The soluble fraction was incubated for 1.5 h rotating at 4 °C with Strep-Tactin agarose beads (IBA Lifesciences, 2-1201-010) to immunoprecipitate (IP) over-expressed SAN1ssf, or with rabbit polyclonal anti-FAM120b antibody (1:1000, Abcam ab106455) and protein A/G Sepharose (Santa Cruz, SC-2003) to precipitate endogenous SAN1. Samples were washed 3× for 5 min in 1× Co-IP buffer, and eluted with 10 mM desthiobiotin for 2 h at 4 °C (SAN1-ssf), or by boiling samples in 4× Laemmli sample buffer. Co-IP samples and inputs were analyzed by immunoblot with SAN1(1:1000, Abcam ab106455), SETX (1:1000, Novus Biologicals NB100-57542), FLAG (1:1000, Sigma F1084), GFP (1:500, Abcam Ab13970), and Phospho-Chk2 Thr68 (1:500, Cell Signaling Tech 2661) antibodies. 3% milk was used for blocking.

**Co-IP of SAN1-FLAG and myc-SETX-SID fragment**. 293T cells were co-transfected using calcium phosphate with either SAN1-FLAG, SAN1ΔRep-FLAG, and myc-SETX-SID fragment, and lysed in buffer containing 25 mM HEPES pH 7.4, 150 mM NaCl, 0.5% Triton-X 100, 0.5 mM EDTA, 1 mM MgCl$_2$, 2 mM DTT, 1 mM PMSF, 10 μg/mL leupeptin, 20 μg/mL aprotinin 24 h later. Cell lysates were then incubated with anti-FLAG M2 agarose for 1 h at 4 °C rotating before three washes. Samples were boiled for 8 min in 4× LSB and analyzed by immunoblot for FLAG (Sigma F1084) and (Myc 9E10, non-commercial) antibodies at 1:1000.

**Cell cycle analysis**. HeLa cells were transfected with indicated siRNA and then treated with vehicle or 5 μM Cisplatin for 2 h, replaced with fresh media and 24 h later harvested for flow cytometric analysis. Cells were washed in PBS and then fixed in ice-cold 70% ethanol for at least 2 h. The cells were washed in PBS and resuspended in propidium iodide staining solution (PBS, 0.1% TritonX-100, 0.2 mg/mL DNase-free RNase A (Sigma), 20 μg/mL LPI (Sigma)) and analyzed using a FACSCalibur machine (BD).

**Yeast-two hybrid screen**. The yeast two-hybrid screen using full-length murine SAN1 as bait was carried out by Hybrigenics Corporation, Cambridge, MA using a mouse brain library (Supplementary Figure 5A related to Fig. 7).

**RNA/DNA hybrid slot-blot assay**. Hybrids were detected as described by Sollier et al.[14]. Total nucleic acids were extracted from cells with the Qiagen DNeasy kit. DNA (1 μg) was spotted in duplicate wells onto positively-charged nylon membrane using a slot-blot apparatus, cross linked by UV treatment, and one well was probed with the S9.6 antibody (1:1000, Kerafast ENH001). The DNA in the duplicate well was denatured for 10 min in 0.5 N NaOH, 1.5 M NaCl, then neutralized for 10 min in 1 M NaCl, 0.5 M Tris-HCl (pH 7.0) and probed with an antibody against ssDNA (1:10000, EMD Millipore MAB3034). Spots were detected with an anti-HRP secondary anti-mouse antibody (1:5000, Jackson ImmunoResearch 115-035-174), and imaged and quantified with an Amersham Imager 600.

**Data availability**. All relevant original data are available from the authors on reasonable request.

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

## Acknowledgements

This work was supported by grants to IGM (R01 GM050526 and R35 CA197571). We would like to thank Steven West (Cancer Research UK) for the Senataxin-FLAG-GFP HeLa cell line, Agata Smogorzewska for the FAN1 antibody, and David Cortez (Vanderbilt University) for reading the manuscript.

## Author contributions

A.M.A., H.J.M. and T.M.E. designed and performed experiments and analyzed data; I.G.M. designed experiments and interpreted data; A.D. provided the mitotic spread data for analysis of radials.

## Additional information

**Competing interests:** The authors declare no competing interests.

