## [Peer Review File · Nature Communications]

Reviewers' comments:

Reviewer #1 (Remarks to the Author):

Andrews et al identified a gene, previously described as a transcriptional coactivator, as a 5' single strand DNA exonuclease, related to the FEN1 family of structure specific nucleases. Knockout of the gene resulted in increased cellular sensitivity to MMC and cisplatin, compounds that can form, among other adducts, interstrand crosslinks (ICLs). They termed the protein SAN1 because it binds the RNA: DNA helicase Senataxin, an interaction that was enhanced by exposure of cells to MMC. Furthermore, DNA damage, marked by γ -H2AX intensity, was elevated in SAN1 deficient cells, relative to wild type cells, following exposure to MMC. SAN1^{-/-} cells also showed an increase in radial chromosomes and chromosomal aberrations after MMC treatment, as compared to wild type cells. R loop levels were increased in SAN1^{-/-} cells after MMC treatment. They suggested that unresolved R loops were responsible for the elevated DNA damage, chromosomal aberrations, and increased sensitivity to MMC and cisplatin. A transcriptional basis for the defects induced by the loss of SAN1 activity was further supported by the demonstration that incubation of SAN1^{-/-} cells with an RNA polymerase inhibitor lowered the γ -H2AX intensity levels to those of wild type cells. They propose that MMC ICLs, by blocking RNA polymerase, provoke R loop formation. The R loops, and the associated ICLs, would be resolved by the action of Senataxin and SAN1. Absent SAN1, R loops would be more persistent, providing targets for cleavage activities, resulting in enhanced DNA damage.

Comments and suggestions

1. The authors describe a series of biochemical experiments in which they characterize SAN1 as an exonuclease, releasing 3-7 nt fragments from single stranded DNA, while inactive on double stranded substrates (Fig 1, 2). Cleavage of the single strand DNA required a minimum length of 25 nucleotides, presumably a reflection of the size of the enzyme. Splayed arm substrates were also tested, some with duplex arms. The enzyme was most active on a substrate with both arms single stranded, showed much reduced activity on a substrate with a duplex on the non-digested arm, and no activity on a substrate with two duplex arms. These experiments are straightforward. However, the substrate that would seem to most closely match the author's scenario for the involvement of SAN1 in ICL repair was that with poor activity- the fork with the non-digested 3' arm in double stranded form. Some consideration of this issue would be appropriate, including the question of whether the substrate digestion preference reflects substrate binding affinity or activity once bound. Also, directly relevant to the authors' model, it would be of interest to know if the activity changed on a substrate with an RNA: DNA hybrid on the non-digested arm.

2. The authors propose that SAN1 be considered another nuclease involved in ICL repair. The exonucleases previously identified as having important roles in ICL repair-SNM1A and FAN1- can digest past an ICL, resulting in unhooking of the lesion. The biochemical data presented here suggest that SAN1 cannot do this. Thus, as noted in the Discussion, a partner nuclease would be required. SNM1A was tested in Fig 6E, what about FAN1? Furthermore, there is no evidence that SAN1 actually contributes to ICL unhooking in vivo. A demonstration that SAN1 was important for release of ICLs, perhaps by alkaline comet analysis, would greatly strengthen the proposal.

3. The authors tested the sensitivity of SAN1 knockout cells to various DNA damaging agents (Fig 3, S3). While there was no increased sensitivity to ionizing radiation, hydroxyurea, camptothecin, or MMS, there was reduced survival, relative to wild type, when cells were exposed to cisplatin, MMC, or etoposide. Although they dismiss the effect of etoposide, the highly significant decline in survival at the highest concentration was greater than that seen with MMC (Fig S3). Since etoposide attacks Topoisomerase 2 α and β , and trapped topoisomerase blocks transcription (Genes 7: piiE32), SAN1 may be relevant to the cellular response to any treatment that increases R loops. It would be informative to ask about the influence of SAN1 loss on the sensitivity to UV, which can block transcription, is well established as a substrate for transcription coupled repair (TC-NER), and can enhance R loop frequency. Additionally, knockdown of the RNA: DNA helicase Aquarius

(Mol Cell 56: 777) would increase R loops without direct DNA damage. These experiments would address the possibility that the SAN1 is important for R loop resolution, regardless of the initiator of their formation.

4. Treatment of SAN1^{-/-} cells with MMC increases the frequency of radial chromosomes, a mark of Fanconi Anemia cells. However, when FANCD2 deficiency was combined with SAN1^{-/-} the cells were more sensitive to cis Platin and MMC than the individually deficient cells (Fig 5 E, F). They interpret this as indicating that SAN1 is not epistatic to the FA pathway. There is an interesting feature of the experiment with MMC that should be noted. The MMC concentrations required to show an effect on SAN1^{-/-} cell survival were 10-fold higher than those in the experiments with FANCD2 deficient cells. In the experiment in Fig 5F the MMC concentrations had no effect on SAN1^{-/-} cell survival relative to wt cells. However, in the FANCD2 deficient background the absence of SAN1 did further sensitize the cells to the low concentrations of MMC. This implies that the FA pathway dominates the response to MMC and masks the loss of SAN1 at the lower concentrations of the drug. Presumably, at higher drug concentrations the FA pathway is overwhelmed and additional options, among them SAN1, contribute to survival.

5. Examination of the intensity of γ -H2AX staining in cells with or without SAN1 and with or without MMC exposure clearly shows an increase in signal in the absence of SAN1, quite strikingly in the case of SAN1^{-/-} cells treated with MMC (Fig 5C, D). This experiment presents data from cells not exposed to MMC, indicating that an increase in γ -H2AX signal in SAN1^{-/-} cells is not absolutely dependent on MMC (see above). A key experiment in support of the authors' model is their demonstration that cordycepin treatment suppresses the increase in MMC induced γ -H2AX staining in the SAN1^{-/-} cells (Fig 6A, B). It would be helpful to present the results from the control experiments- wt and SAN1^{-/-} cells w/o cordycepin, no MMC. This would be consistent with the data set shown in Fig 5. It would also be important to repeat the experiment with other inhibitors of RNA polymerase since cordycepin has many targets (Life Sci 2013, 93: 863).

6. The interaction of SAN1 and SETX is shown by IP in Fig 7A. Fig 7B indicates a substantial increase in the SETX band in nucleoplasm from cells treated with MMC. The text states that there was a decline in SETX-SAN1 in the chromatin fraction, as marked by LaminB1. However, the chromatin fraction results are not apparent in this panel, and no LaminB1 is indicated. The text and the figure need to be reconciled.

The very strong increase in the SETX-GFP band from the MMC treated sample in 7B does not appear to be matched by a similar increase in the results shown in Fig 7C. Is this because the SETX is the endogenous protein (it is not labeled "GFP")? If this is the case, then the increased association of endogenous SETX with SAN1 following MMC treatment may be much more modest than suggested by the result with the GFP tagged protein in Fig 7B.

The scales of the survival curves in Fig 7 are not consistent. It would be easier to evaluate if they were on all the same scale. Fig 7J appears to reiterate the point made indirectly in Fig 5- that in the absence of SAN1 there is an increase in R loops without requirement for MMC treatment.

7. In the Discussion the authors propose a model in which R loops formed by blocks to transcription by MMC are resolved by the action of SETX and SAN1. They suggest that the requirement for a 5' single strand end as a substrate for SAN1 would be met by the introduction of breaks in the single strand side of the R loop (consistent with earlier work from other groups, JBC 275: 24163). It has been shown that TC-NER nucleases promote cleavages of R loops (Mol Cell 2014, 56: 777). This raises the obvious question of whether XPG and/or XPF are epistatic with SAN1 in the assays presented here. These experiments should be done. Furthermore, the defining experimental test of R loops in vivo is the sensitivity to RNaseH1 overexpression. Transfection of a plasmid expressing RNaseH1 has become a standard control experiment, and should be applied to the relevant biological experiments in this study. The model predicts that, without associated R loops, replication independent ICL repair would decline and survival following MMC treatment

would be further reduced.

8. In the immunofluorescence experiments cells were treated for lengthy periods (24, 30 hrs) with MMC, with no indication of the cell cycle distribution of the population at the time of immunostaining. It is possible that the cells have stalled in S phase. The model of replication independent repair of ICLs would be strengthened by eliminating this possibility by showing the distribution of cells after the long treatment, and demonstrating the R loop/ γ -H2AX dynamics in G1 cells. This can be done by including an antibody against a cell cycle marker in the immunofluorescence assays.

General comment: This report makes a plausible argument for the linkage of R loop resolution and some level of resistance to ICL forming agents. The idea of a double lesion (R loop/ICL) attracting functions to remove the one, with the fortuitous resolution of the other, is interesting and intriguing. This is particularly relevant for ICLs formed by MMC, which introduce relatively little helix distortion, and would not be detected by global NER. Thus the study provides an expanded view of replication independent responses of cells to DNA lesions. This aspect might be emphasized in the title, rather than the claim of "efficient repair of ICLs" for which there is no direct evidence.

Reviewer #2 (Remarks to the Author):

The work by Andrews and colleagues uncovers that the previously uncharacterized protein FAM120B (named SAN1 by the authors) is an exonuclease involved in ICL repair. An extensive biochemical analysis with recombinant SAN1 protein and numerous DNA substrates reveals a 5' exonuclease activity targeting single-stranded (ss)DNA, and a preference for splayed arm substrates. Depletion of SAN1 from HeLa cells and MEFs results in an increased sensitivity toward cross-linking agents, indicating a potential role in ICL repair. This phenotype is rescued by the expression of SAN1 WT protein but not a catalytically inactive mutant. Furthermore, the authors suggest that SAN1 acts independently of the FA pathway (i.e. not in a replication-coupled pathway) based on the non-epistatic relationship with FANCD2. Finally, a direct interaction between SAN1 and the RNA-DNA helicase SETX is described, which is important for SAN1 function. SETX is involved in the removal of R-loops upon transcription stalling. The authors therefore examine the impact of SAN1 loss on R-loop levels and DNA damage, and find that R-loops accumulate in SAN1 $-/-$ cells after exposure to cross-linking agents. Since DNA damage appears to be dependent on transcription, the authors conclude that SAN1 together with SETX participate in a novel transcription-coupled ICL repair mechanism.

The data presented contains interesting aspects - especially the biochemistry is convincing - yet other parts suffer from a lack of clarity and a questionable experimental design with the use of cordycepin as inhibitor of transcription (see major comment 4). It is clear from the data that SAN1 is somehow involved in ICL repair, but basic questions about its role as well as the overall proposed mechanism of SAN1 action remain unclear (partly due to the lack of a model figure). Why does SETX not recruit SAN1 to every R-loop (i.e. how does SAN1 'know' on which R-loops to act)? Where does the initial incision in the ssDNA of the R-loop come from? Where does the specificity for ICLs come from (RNA pol II stalls at various lesions)? Would RNA pol II need to be removed for SAN1 to resect the DNA to the ICL? The conclusion that SAN1 functions in a transcription-dependent pathway is not well-supported. What is the epistatic relationship between SAN1 and known transcription-coupled repair factors such as CSA and CSB? Addressing these relationships would provide more compelling evidence for placing SAN1 in a transcription-dependent pathway.

Overall, this manuscript contains much useful information, but the connection to transcription is dubious. Therefore, in addition to addressing the other points below, the authors should either

remove the transcription-related data and modify their conclusions accordingly, or strengthen this part with the recommended additional experiments.

Major comments:

1) The conclusion that SAN1 releases 3-7 nt long fragments appears to be solely based on Fig. 1H. Why is this cleavage pattern not observed in the other assays? Figs. 1C, 1J, 2B, 2C, S1E, and S2 show only single released products, and their respective sizes are difficult to estimate on the gel. It is not clear how often these experiments have been performed and therefore how robust they are. Moreover, a time-course assay would be more appropriate and informative to investigate the preference of SAN1 between different substrates (e.g. splayed vs. flap arm structures) than an end point assay.

2) There is a discrepancy between the emphasized preference of SAN1 for splayed arm DNA structures observed *in vitro* and the suggested action of SAN1 on R-loops. The authors state in line 155 that "a splayed arm structure appears to be important because cleavage of a 5' flap is less efficient when the 3' arm is double-stranded". If the ssDNA strand in an R-loop was incised by an endonuclease, wouldn't the resulting structure be a flap (due to the RNA-DNA double-strand) rather than a splayed arm, i.e. a non-preferred substrate for SAN1?

3) Why are the non-repaired ICLs in SAN1 *-/-* cells not repaired by the FA pathway? Do the cross-linking agents employed induce other DNA lesions that might contribute to the observed phenotypes?

4) To inhibit transcription and assess if SAN1 functions in a transcription-coupled mechanism, the authors treated HeLa cells with cordycepin, an adenosine derivative. The choice of this compound is highly questionable, as it is not a specific inhibitor of RNA pol II (it hardly incorporates cordycepin, if at all). In fact, cordycepin is used more frequently to inhibit polyadenylation because poly(A) polymerase readily incorporates it. The resulting mRNAs are destabilized, causing an apparent transcription defect, which is different from the transcription inhibition the authors intended. Furthermore, cordycepin targets various ATP-dependent processes including cell signaling, all of which could contribute to the observed effects. In order to convince this reviewer that the results are indeed based on inhibited transcription, the authors need to repeat the experiments with a specific RNA pol II inhibitor (e.g. alpha-amanitin).

5) The reduction of gammaH2AX foci in SAN1 *-/-* cells after cordycepin treatment only suggests that transcription - assuming it was inhibited and no other process was impaired - induces DNA damage. However, this experiment cannot provide any mechanistic insights into SAN1 function. DNA damage upon transcription could be a cause or a consequence of the SAN1 deficiency. Either the lack of SAN1 results in increased ICL levels (e.g. if it acts independently of transcription) that lead to increased transcription stalling with R-loop formation and subsequent DNA damage, and/or transcription stalling causes DNA damage because SAN1 deficient cells cannot efficiently resolve such situations. Therefore, the provided data does not support the involvement of SAN1 in a transcription-coupled ICL repair pathway. As outlined above, the authors should instead focus on the epistatic relationship of SAN1 with known transcription-coupled repair proteins (i.e. CSA and CSB).

6) The main text (lines 314-316) describes that upon MMC treatment the SETX-SAN1 interaction was increased in the nucleoplasm but decreased in the chromatin fraction, while Fig. 7B does not show any chromatin fraction, and LaminB1 described in the text is missing in the figure as well. Shouldn't SAN1 and SETX interact on the chromatin (where R-loops are) after MMC treatment?

7) The pulldown assay in Fig. 7C is not convincing. Lane 4 shows the pulldown of a supposedly elevated level of SETX but also shows an increased amount of SAN1. These bands need to be quantified and normalized. How often has this been repeated? Is this data significant? Also, was there really no MMC in the last lane when cordycepin was added?

8) R-loop levels in cells were tested by both immunofluorescence (Fig. 7J) and a dot blot assay (Fig. 7K), but the results are inconsistent and very confusing. MMC treatment in HeLa WT cells lead to no increase in R-loop levels in Fig. 7J, yet a relatively strong 1.4-fold increase was detected in Fig. 7K (compared to a 1.8-fold increase in SAN1 *-/-*). In contrast, "SAN1 *-/-* + WT" (WT here refers to wild-type SAN1 protein expressed in SAN1 *-/-* cells) shows the same hybrid level as HeLa in Fig. 7J as expected (i.e. the SAN1 rescued cells behave like HeLa WT), while they do not show

1.4-fold increased hybrid levels after MMC treatment (compare panels 1-4). How can this be explained? Furthermore, the slight increase in R-loop levels without MMC treatment in SAN1 $-/-$ cells as compared to HeLa WT in Fig. 7J was not detected in Fig. 7K.

Minor comments:

- 1) Fig. 3D - the bars indicating statistical significance partly overlap with the last data points. Please move for clarity.
- 2) Fig. 7E - y-axis is not in log-scale, while all other y-axes are.
- 3) Supplementary Fig. 5 appears to be missing and could not be reviewed.
- 4) The manuscript would greatly benefit from a model figure so the reader can grasp the mechanism that the authors propose.

Response to Reviewer comments:

We thank the reviewers for their helpful, constructive comments, which have helped to significantly strengthen our manuscript. We have performed multiple additional experiments to address several of the points they raised, removed the cordycepin data, added new data on 53BP1 foci (new Figure 5E), and changed the text to avoid the conclusion that SAN1 acts specifically in transcription-coupled repair. The point-by-point response given below details the major revisions to the manuscript.

Reviewer #1 (Remarks to the Author):

Andrews et al identified a gene, previously described as a transcriptional coactivator, as a 5' single strand DNA exonuclease, related to the FEN1 family of structure specific nucleases. Knockout of the gene resulted in increased cellular sensitivity to MMC and cisplatin, compounds that can form, among other adducts, interstrand crosslinks (ICLs). They termed the protein SAN1 because it binds the RNA: DNA helicase Senataxin, an interaction that was enhanced by exposure of cells to MMC. Furthermore, DNA damage, marked by γ -H2AX intensity, was elevated in SAN1 deficient cells, relative to wild type cells, following exposure to MMC. SAN1^{-/-} cells also showed an increase in radial chromosomes and chromosomal aberrations after MMC treatment, as compared to wild type cells. R loop levels were increased in SAN1^{-/-} cells after MMC treatment. They suggested that unresolved R loops were responsible for the elevated DNA damage, chromosomal aberrations, and increased sensitivity to MMC and cisplatin. A transcriptional basis for the defects induced by the loss of SAN1 activity was further supported by the demonstration that incubation of SAN1^{-/-} cells with an RNA polymerase inhibitor lowered the γ -H2AX intensity levels to those of wild type cells. They propose that MMC ICLs, by blocking RNA polymerase, provoke R loop formation. The R loops, and the associated ICLs, would be resolved by the action of Senataxin and SAN1. Absent SAN1, R loops would be more persistent, providing targets for cleavage activities, resulting in enhanced DNA damage.

Comments and suggestions

1. The authors describe a series of biochemical experiments in which they characterize SAN1 as an exonuclease, releasing 3-7 nt fragments from single stranded DNA, while inactive on double stranded substrates (Fig 1, 2). Cleavage of the single strand DNA required a minimum length of 25 nucleotides, presumably a reflection of the size of the enzyme. Splayed arm substrates were also tested, some with duplex arms. The enzyme was most active on a substrate with both arms single stranded, showed much reduced activity on a substrate with a duplex on the non-digested arm, and no activity on a substrate with two duplex arms. These experiments are straightforward. However, the substrate that would seem to most closely match the author's scenario for the involvement of SAN1 in ICL repair was that with poor activity- the fork with the non-digested 3' arm in double stranded form. Some consideration of this issue would be appropriate, including the question of whether the substrate digestion preference reflects substrate binding affinity or activity once bound. Also, directly relevant to the authors' model, it would be of interest to know if the activity changed on a substrate with an RNA: DNA hybrid on the non-digested arm.

- While the R-loop structure would initially resemble the flap structure (with a 3' duplex arm) Senataxin is expected to unwind the RNA/DNA hybrid, producing a structure more closely resembling the splayed arm than a fork with an RNA:DNA duplex arm. We now include in Supplementary Figure 7 a speculative model (as suggested by both reviewers) that outlines this idea.

- With regard to the question of whether the preference of SAN1 for a splayed arm versus fork structure reflects an affinity difference or a difference in k_{cat} , we propose a more detailed biochemical model (new Figure 2F) in which the nuclease domain binds to and

cleaves the free 5' end while a second binding site interacts with the substrate 3' to this. This type of model can explain the inability of SAN1 to cleave ssDNA 25 nt or shorter in length (because such a ssDNA would be too small to bind both sites) and why it can act efficiently on a splayed arm that is shorter than 25 nt (because the second site can bind the bottom strand of the DNA substrate). It would act much less efficiently on a fork substrate because the second site could not bind the duplex arm of the fork.

- We feel that a thorough and detailed investigation of this model is beyond the scope of this manuscript, but we performed an EMSA assay to compare the binding of the SAN1 D90A mutant (catalytically inactive) to either a 5' ³²P splayed arm structure or a 5' fork with a duplex arm. As shown below, in this preliminary experiment we see a small shift in Lane 2 (splayed arm structure plus SAN1 DA), but do not see any shift in Lane 4 (5' flap structure plus SAN1 DA). This result would be consistent with the splayed arm having a higher affinity for SAN1 than the fork. However, much further work, including kinetic assays, would be needed to definitively prove this to be the case

2. The authors propose that SAN1 be considered another nuclease involved in ICL repair. The exonucleases previously identified as having important roles in ICL repair-SNM1A and FAN1- can digest past an ICL, resulting in unhooking of the lesion. The biochemical data presented here suggest that SAN1 cannot do this. Thus, as noted in the Discussion, a partner nuclease would be required. SNM1A was tested in Fig 6E, what about FAN1? Furthermore, there is no evidence that SAN1 actually contributes to ICL unhooking in vivo. A demonstration that SAN1 was important for release of ICLs, perhaps by alkaline comet analysis, would greatly strengthen the proposal.

- As suggested, we have now tested the effect of silencing FAN1 in WT and SAN1^{-/-} HeLa cells. FAN1 depletion increases sensitivity to MMC by about the same degree as loss of SAN1, but there is no additive effect, suggesting that FAN1 is epistatic with SAN1 (new Supplementary Figure S4f-g). We do not yet know the functional relationship between these two nucleases – and note that even after 8 years it remains unclear exactly what the function of FAN1 is in ICL repair – but we can speculate that SNM1A or FAN1 can participate in unhooking of the lesion after the initial processing of the free 5' end of the DNA by SAN1.

- We have made several attempts to use the comet assay to examine release of ICLs, as suggested, but unfortunately the assay is not sufficiently sensitive for the relatively small impact of the SAN1 deletion to produce a clear, quantifiable comet tail in our hands

3. The authors tested the sensitivity of SAN1 knockout cells to various DNA damaging agents (Fig 3, S3). While there was no increased sensitivity to ionizing radiation, hydroxyurea, camptothecin, or MMS, there was reduced survival, relative to wild type, when cells were exposed to cisplatin, MMC, or etoposide. Although they dismiss the effect of etoposide, the highly significant decline in survival at the highest concentration was greater than that seen with MMC (Fig S3). Since etoposide attacks Topoisomerase 2α and β, and trapped topoisomerase blocks transcription (Genes 7: piiE32), SAN1 may be relevant to the cellular response to any treatment that increases R loops. It would be

informative to ask about the influence of SAN1 loss on the sensitivity to UV, which can block transcription, is well established as a substrate for transcription coupled repair (TC-NER), and can enhance R loop frequency. Additionally, knockdown of the RNA: DNA helicase Aquarius (Mol Cell 56: 777) would increase R loops without direct DNA damage. These experiments would address the possibility that the SAN1 is important for R loop resolution, regardless of the initiator of their formation.

- We have added the UV assay as suggested, but see no increased sensitivity in SAN1^{-/-} cells (new Supplementary Figure 6e). We agree that the SAN1^{-/-} cells display a survival defect in response to etoposide, and it is possible that SAN1 might be required for the repair of etoposide-induced lesions through an alternative repair process, however we believe it is unlikely that this is a result of the increased R-loops. This is largely because we observe no survival defect in response to Camptothecin or UV, two compounds that also block transcription and can induce R-loop formation. Because we cannot provide a strong rationale or cause for the etoposide sensitivity at a high concentration, we have removed these data from the new version of the manuscript.

- Although we have not silenced Aquarius, we did silence Senataxin, which also has helicase activity and is needed for processing of R-loops (Supplementary Figure 5). Importantly, we want to emphasize that we are not suggesting that San1 is necessary for R-loop resolution, only that it is involved in the response to ICLs that generate adjacent R-loops, likely through its interaction with Senataxin (New Supplementary Figure S7).

- While it is possible that the DNA damage present in SAN1^{-/-} in response to ICLs is partially attributable to R-loop formation, the presence of radial chromosomes and aberrations is a specific form of DNA damage that results from unrepaired ICLs (Deans and West et al., 2011). Moreover the genomic instability associated with increased R-loop levels has not been shown to result in radial chromosomes or aberrations, as they result from collapsed replication forks that generate one-sided double-stranded breaks. For these reasons we believe it is unlikely that SAN1 plays a role in general R-loop resolution or the response to treatment that can induce R-loop formation.

4. Treatment of SAN1^{-/-} cells with MMC increases the frequency of radial chromosomes, a mark of Fanconi Anemia cells. However, when FANCD2 deficiency was combined with SAN1^{-/-} the cells were more sensitive to cisplatin and MMC than the individually deficient cells (Fig 5 E, F). They interpret this as indicating that SAN1 is not epistatic to the FA pathway. There is an interesting feature of the experiment with MMC that should be noted. The MMC concentrations required to show an effect on SAN1^{-/-} cell survival were 10-fold higher than those in the experiments with FANCD2 deficient cells. In the experiment in Fig 5F the MMC concentrations had no effect on SAN1^{-/-} cell survival relative to wt cells. However, in the FANCD2 deficient background the absence of SAN1 did further sensitize the cells to the low concentrations of MMC. This implies that the FA pathway dominates the response to MMC and masks the loss of SAN1 at the lower concentrations of the drug. Presumably, at higher drug concentrations the FA pathway is overwhelmed and additional options, among them SAN1, contribute to survival.

- This is a very interesting point brought up by the reviewer, which we agree with and now mention in the results and discussion sections of the manuscript. Additionally, we observe a similar effect upon depletion of the nuclease XPF (FANCD1) in SAN1^{-/-} cells, data that we have now added (New Supplementary figure S4a-c), which further supports this idea.

5. Examination of the intensity of γ -H2AX staining in cells with or without SAN1 and with or without MMC exposure clearly shows an increase in signal in the absence of SAN1, quite strikingly in the case of SAN1^{-/-} cells treated with MMC (Fig 5C, D). This experiment presents data from cells not exposed to MMC, indicating that an increase in γ -H2AX signal in SAN1^{-/-} cells is not absolutely dependent on MMC (see above).

A key experiment in support of the authors' model is their demonstration that cordycepin treatment suppresses the increase in MMC induced γ -H2AX staining in the SAN1^{-/-} cells (Fig 6A, B). It would be helpful to present the results from the control experiments- wt and SAN1^{-/-} cells w/o cordycepin, no MMC. This would be consistent with the data set shown in Fig 5. It would also be important to repeat the experiment with other inhibitors of RNA polymerase since cordycepin has many targets (Life Sci 2013, 93: 863).

- As the reviewer notes, cordycepin is not specific for inhibition of RNA polymerase. Therefore, as was suggested by reviewer #2, we have removed the cordycepin data from the manuscript. We repeated the experiments using alpha-amanitin. However, as reviewer #2 points out, these experiments are very difficult to interpret, because the DNA damage during transcription could be upstream or downstream of SAN1 function. In addition, blocking transcription even for a few hours will alter the expression level of hundreds of proteins, including cell cycle proteins and other DNA repair factors, and extended treatment will cause cell cycle arrest and death. Our experiments with alpha-amanitin for 6 hours gave less convincing results than with cordycepin, but there is a small increase in γ -H2AX positive cells with MMC that is reduced by alpha-amanitin treatment both in the control and SAN1^{-/-} cells, now shown in the new Supplementary Figure 6. However, from these data we cannot determine whether or not SAN1 participates specifically in a transcription-dependent response to ICLs, and we have now modified the text to reflect this point.

6. The interaction of SAN1 and SETX is shown by IP in Fig 7A. Fig 7B indicates a substantial increase in the SETX band in nucleoplasm from cells treated with MMC. The text states that there was a decline in SETX-SAN1 in the chromatin fraction, as marked by LaminB1. However, the chromatin fraction results are not apparent in this panel, and no LaminB1 is indicated. The text and the figure need to be reconciled.

- We apologize for this mistake, left over from an earlier version of the manuscript, and have now corrected the text.

The very strong increase in the SETX-GFP band from the MMC treated sample in 7B does not appear to be matched by a similar increase in the results shown in Fig 7C. Is this because the SETX is the endogenous protein (it is not labeled "GFP")? If this is the case, then the increased association of endogenous SETX with SAN1 following MMC treatment may be much more modest than suggested by the result with the GFP tagged protein in Fig 7B.

- We have now deleted figure 7C because of the issue with the non-specific effects of cordycepin as brought up by the reviewer and as we discussed above in response to comment #5.

The scales of the survival curves in Fig 7 are not consistent. It would be easier to evaluate if they were on all the same scale. Fig 7J appears to reiterate the point made indirectly in Fig 5- that in the absence of SAN1 there is an increase in R loops without requirement for MMC treatment.

-The linear scale was used for Figure 7E because at the highest concentration of MMC used there were zero colonies in some conditions, which cannot be represented in a semi-log format.

- We re-analyzed the data from the immunofluorescence R-loop assay, and performed repeat experiments of the dot blot assay. For the IF assay in Fig 7I we now show the means for each biological replicate, rather than each individual nucleus for every experiment. There is also a clear difference in the R-loop levels and in the response of the HeLa WT and SAN1^{-/-} cells to MMC, which is now indicated by in Figure 7I. Although there is a small increase in R-loops in the SAN1^{-/-} cells in the absence of MMC, this was not detected by the dot blot assay. We do not know the reason for this discrepancy but one is a single cell IF assay, while the other is a bulk assay based on genomic DNA isolation, so the dynamic range, variance, and other parameters will be quite different.

In the Discussion the authors propose a model in which R loops formed by blocks to transcription by MMC are resolved by the action of SETX and SAN1. They suggest that the requirement for a 5' single strand end as a substrate for SAN1 would be met by the introduction of breaks in the single strand side of the R loop (consistent with earlier work from other groups, JBC 275: 24163). It has been shown that TC-NER nucleases promote cleavages of R loops (Mol Cell 2014, 56: 777). This raises the obvious question of whether XPG and/or XPF are epistatic with SAN1 in the assays presented here. These experiments should be done. Furthermore, the defining experimental test of R loops in vivo is the sensitivity to RNaseH1 overexpression. Transfection of a plasmid expressing RNaseH1 has become a standard control experiment, and should be applied to the relevant biological experiments in this study. The model predicts that, without associated R loops, replication independent ICL repair would decline and survival following MMC treatment would be further reduced.

- This is an excellent point. We have tested whether XPF is epistatic with SAN1 at very low concentrations of MMC and cisplatin, and find that it is not: the effects of XPF knockdown is synergistic with loss of SAN1. However, since XPF is involved in the Fanconi pathway, we cannot easily interpret these data. We now mention these results in the Discussion where we also mention the non-epistatic response to silencing FANCD2.

-As the reviewer suggested we have also performed experiments to assess the relationship of SAN1, R-loops, and the cellular response to ICLs. We utilized a RNaseH1-NLS-Mcherry construct to examine the levels of γ H2AX following MMC treatment (new Supplementary figure S6c-d). Due to the complex nature of R-loops as a source of genomic instability, as well as the potential requirement of R-loops for SAN1 to act in ICL repair, the results are somewhat difficult to interpret.

-Consistent with the reviewer's prediction that loss of R-loops would reduce ICL repair efficiency, we observed an increase in the mean fluorescence intensity of γ H2AX in HeLa WT cells over-expressing RNaseH1, following MMC treatment. In contrast, SAN1^{-/-} over-expressing RNaseH1 displayed reduced levels of γ H2AX after MMC treatment. This result suggests that part of the MMC-dependent DNA damage in SAN1^{-/-} cells might be attributable to associated R-loops. Additionally these results are in line with our speculative model that we have now included (new Supplementary Figure S7), in which the presence of R-loops adjacent to an ICL precedes the function of SAN1 in response to ICLs.

In the immunofluorescence experiments cells were treated for lengthy periods (24, 30 hrs) with MMC, with no indication of the cell cycle distribution of the population at the time of immunostaining. It is possible that the cells have stalled in S phase. The model of replication independent repair of ICLs would be strengthened by eliminating this possibility by showing the distribution of cells after the long treatment, and demonstrating the R loop/ γ -H2AX dynamics in G1 cells. This can be done by including an antibody against a cell cycle marker in the immunofluorescence assays.

- To determine if the cells might be stalling during the lengthy periods of MMC treatment in the immunofluorescence experiments, we performed labeling and staining with BRDU. We observed significantly less incorporation of the BRDU in the MMC treated cells (data not shown), indicating that, as the reviewer suggests, the cells are likely to be at least partly stalling in S phase. Due to this result we cannot definitively conclude that SAN1 functions in a totally replication independent manner, only that it acts independently of the Fanconi Anemia pathway from the epistasis experiments with FANCD2 and XPF. We have changed the text to reflect this point.

General comment: This report makes a plausible argument for the linkage of R loop resolution and some level of resistance to ICL forming agents. The idea of a double lesion (R loop/ICL) attracting functions to remove the one, with the fortuitous resolution of the other, is interesting and intriguing. This is particularly relevant for ICLs formed by MMC, which introduce relatively little helix distortion, and would not be detected by global NER. Thus the study provides an expanded view of replication independent responses of cells to DNA lesions. This aspect might be emphasized in the title, rather than the claim of "efficient repair of ICLs" for which there is no direct evidence.

- We agree and have changed the title to read: "**A Senataxin-Associated Exonuclease, SAN1, is Required for Resistance to DNA Interstrand Cross-links**"

Reviewer #2 (Remarks to the Author):

The work by Andrews and colleagues uncovers that the previously uncharacterized protein FAM120B (named SAN1 by the authors) is an exonuclease involved in ICL repair. An extensive biochemical analysis with recombinant SAN1 protein and numerous DNA substrates reveals a 5' exonuclease activity targeting single-stranded (ss)DNA, and a preference for splayed arm substrates. Depletion of SAN1 from HeLa cells and MEFs results in an increased sensitivity toward cross-linking agents, indicating a potential role in ICL repair. This phenotype is rescued by the expression of SAN1 WT protein but not a catalytically inactive mutant. Furthermore, the authors suggest that SAN1 acts independently of the FA pathway (i.e. not in a replication-coupled pathway) based on the non-epistatic relationship with FANCD2. Finally, a direct interaction between SAN1 and the RNA-DNA helicase SETX is described, which is important for SAN1 function. SETX is involved in the removal of R-loops upon transcription stalling. The authors therefore examine the impact of SAN1 loss on R-loop levels and DNA damage, and find that R-loops accumulate in SAN1 -/- cells after exposure to cross-linking agents. Since DNA damage appears to be dependent on transcription, the authors conclude that SAN1 together with SETX participate in a novel transcription-coupled ICL repair mechanism.

The data presented contains interesting aspects - especially the biochemistry is convincing - yet other parts suffer from a lack of clarity and a questionable experimental design with the use of cordycepin as inhibitor of transcription (see major comment 4). It is clear from the data that SAN1 is somehow involved in ICL repair, but basic questions about its role as well as the overall proposed mechanism of SAN1 action remain unclear (partly due to the lack of a model figure).

Why does SETX not recruit SAN1 to every R-loop (i.e. how does SAN1 'know' on which R-loops to act)? Where does the initial incision in the ssDNA of the R-loop come from? Where does the specificity for ICLs come from (RNA pol II stalls at various lesions)? Would RNA pol II need to be removed for SAN1 to resect the DNA to the ICL? The conclusion that SAN1 functions in a transcription-dependent pathway is not well-supported. What is the epistatic relationship between SAN1 and known transcription-coupled repair factors such as CSA and CSB? Addressing these relationships would provide more compelling evidence for placing SAN1 in a transcription-dependent pathway.

Overall, this manuscript contains much useful information, but the connection to transcription is dubious. Therefore, in addition to addressing the other points below, the authors should either remove the transcription-related data and modify their conclusions accordingly, or strengthen this part with the recommended additional experiments.

Major comments:

1) The conclusion that SAN1 releases 3-7 nt long fragments appears to be solely based on Fig. 1H. Why is this cleavage pattern not observed in the other assays? Figs. 1C, 1J, 2B, 2C, S1E, and S2 show only single released products, and their respective sizes are difficult to estimate on the gel. It is not clear how often these experiments have been performed and therefore how robust they are. Moreover, a time-course assay would be more appropriate and informative to investigate the preference of SAN1 between different substrates (e.g. splayed vs. flap arm structures) than an end point assay.

- We have added another example of the formation of the ~3 – 7nt fragments from a splayed arm structure, in new Supplementary Figure 1F. The smaller fragment is also just visible in Figure S1C. The relative intensities of the two fragments varies with the sequence and the structure of the substrates, for unknown reasons.

- We showed a time course for the 5' labeled ssDNA substrate in Figure S1H. All assays were performed under initial rate conditions, so we do not believe that time courses would add further information. However, we do now include a time course for 5' labeled ssDNA digestion in new Supplementary Figure 2b.

2) *There is a discrepancy between the emphasized preference of SAN1 for splayed arm DNA structures observed in vitro and the suggested action of SAN1 on R-loops. The authors state in line 155 that "a splayed arm structure appears to be important because cleavage of a 5' flap is less efficient when the 3' arm is double-stranded". If the ssDNA strand in an R-loop was incised by an endonuclease, wouldn't the resulting structure be a flap (due to the RNA-DNA double-strand) rather than a splayed arm, i.e. a non-preferred substrate for SAN1?*

- While the R-loop structure would initially resemble the flap structure (with a 3' duplex arm) Senataxin is expected to unwind the RNA/DNA hybrid, producing a structure more closely resembling the splayed arm. We now include a speculative model (as suggested by both reviewers) that outlines this idea (new Supplementary Figure S7).

3) *Why are the non-repaired ICLs in SAN1 -/- cells not repaired by the FA pathway? Do the cross-linking agents employed induce other DNA lesions that might contribute to the observed phenotypes?*

- One possibility is that the ICLs left unrepaired in SAN1^{-/-} cells require repair by FA independent mechanisms, which are poorly defined but exist given the sensitivity of cells depleted of other nucleases that have been shown to function wholly (SNM1a) or partly (FAN1) independently of the FA pathway in ICL repair.

- Alternatively the ICLs in SAN1^{-/-} cells might go unrepaired due to the high levels of ICLs overwhelming of FA pathway. Indeed, as noted by reviewer #1, at very low MMC concentrations the loss of SAN1 has no effect on colony survival, which suggests that it might function in a secondary pathway for conditions under which the FA pathway is unable to repair all ICLs (e.g., high MMC concentrations).

-The toxicities of the cross-linking agents employed have been previously established to be mostly a result of the **inter**strand cross-links that form, and not from the **intra**strand cross-link lesions. Moreover, the presence of radial chromosomes and aberrations in SAN1^{-/-} cells is a form of DNA damage that is specific to cells' inability to repair **inter**strand cross-links, and form from collapsed replication forks that result in one-sided double strand breaks.

4) *To inhibit transcription and assess if SAN1 functions in a transcription-coupled mechanism, the authors treated HeLa cells with cordycepin, an adenosine derivative. The choice of this compound is highly questionable, as it is not a specific inhibitor of RNA pol II (it hardly incorporates cordycepin, if at all). In fact, cordycepin is used more frequently to inhibit polyadenylation because poly(A) polymerase readily incorporates it. The resulting mRNAs are destabilized, causing an apparent transcription defect, which is different from the transcription inhibition the authors intended. Furthermore, cordycepin targets various ATP-dependent processes including cell signaling, all of which could contribute to the observed effects. In order to convince this reviewer that the results are indeed based on inhibited transcription, the authors need to repeat the experiments with a specific RNA pol II inhibitor (e.g. alpha-amanitin).*

- As the reviewer notes, cordycepin is not specific for inhibition of RNA polymerase. Therefore, as was suggested by reviewer #2, we have removed the cordycepin data from the

manuscript and adjusted our conclusions. We conducted similar experiments using alpha-amanitin. However, as reviewer #2 points out, these experiments are very difficult to interpret, because the DNA damage during transcription could be a cause or consequence of loss of SAN1. In addition, blocking transcription even for a few hours will alter the expression level of hundreds of proteins, including cell cycle proteins and other DNA repair factors, and extended treatment will cause cell cycle arrest and death. Our experiments with alpha-amanitin for 6 hours gave less convincing results than with cordycepin, but there is a small increase in gH2AX positive cells with MMC that is reduced by alpha-amanitin treatment both in the control and SAN1^{-/-} cells, now shown in the new Supplementary Figure 6.

5) *The reduction of gammaH2AX foci in SAN1^{-/-} cells after cordycepin treatment only suggests that transcription - assuming it was inhibited and no other process was impaired - induces DNA damage. However, this experiment cannot provide any mechanistic insights into SAN1 function. DNA damage upon transcription could be a cause or a consequence of the SAN1 deficiency. Either the lack of SAN1 results in increased ICL levels (e.g. if it acts independently of transcription) that lead to increased transcription stalling with R-loop formation and subsequent DNA damage, and/or transcription stalling causes DNA damage because SAN1 deficient cells cannot efficiently resolve such situations. Therefore, the provided data does not support the involvement of SAN1 in a transcription-coupled ICL repair pathway. As outlined above, the authors should instead focus on the epistatic relationship of SAN1 with known transcription-coupled repair proteins (i.e. CSA and CSB).*

- We agree with this point. In addition, prolonged inhibition of transcription will result in many changes in gene expression with unknown consequences. Therefore, we have deleted the data related to cordycepin. As the reviewer suggested, we did test the epistatic relationship to CSB. In our HeLa cells silencing of CSB had no effect on colony survival in response to MMC, and did not further reduce survival in SAN1^{-/-} cells. We are including these negative data here, but not in the revised manuscript.

6) *The main text (lines 314-316) describes that upon MMC treatment the SETX-SAN1 interaction was increased in the nucleoplasm but decreased in the chromatin fraction, while Fig. 7B does not show any chromatin fraction, and LaminB1 described in the text is missing in the figure as well. Shouldn't SAN1 and SETX interact on the chromatin (where R-loops are) after MMC treatment?*

- We apologize for this mistake, and have corrected the text. We do not know at this point at which step in ICL processing SAN1 interacts with SETX.

7) The pulldown assay in Fig. 7C is not convincing. Lane 4 shows the pulldown of a supposedly elevated level of SETX but also shows an increased amount of SAN1. These bands need to be quantified and normalized. How often has this been repeated? Is this data significant? Also, was there really no MMC in the last lane when cordycepin was added?

- Because this figure relates to the effects of cordycepin we have deleted it along with the other cordycepin data from the manuscript.

8) R-loop levels in cells were tested by both immunofluorescence (Fig. 7J) and a dot blot assay (Fig. 7K), but the results are inconsistent and very confusing. MMC treatment in HeLa WT cells lead to no increase in R-loop levels in Fig. 7J, yet a relatively strong 1.4-fold increase was detected in Fig. 7K (compared to a 1.8-fold increase in SAN1 -/-). In contrast, "SAN1 -/- + WT" (WT here refers to wild-type SAN1 protein expressed in SAN1 -/- cells) shows the same hybrid level as HeLa in Fig. 7J as expected (i.e. the SAN1 rescued cells behave like HeLa WT), while they do not show 1.4-fold increased hybrid levels after MMC treatment (compare panels 1-4). How can this be explained? Furthermore, the slight increase in R-loop levels without MMC treatment in SAN1 -/- cells as compared to HeLa WT in Fig. 7J was not detected in Fig. 7K.

- We have now repeated the dot blot assay multiple times, and show error bars and statistics (t-test) (now Figure 7J). Overall the results are the same as in the original figure, but there is no significant difference in S9.6 spot intensities between WT and SAN1-/- cells in the absence of MMC. In WT HeLa cells there is a small (20%) increase in response to MMC but this is not significant. The small discrepancy between WT vs SAN1-/- cells in the absence of MMC persists between the two different assays, for unknown reasons, but one is a single cell IF assay, while the other is a bulk assay based on genomic DNA isolation, so the dynamic range, variance, and other parameters will be quite different.

Minor comments:

1) Fig. 3D - the bars indicating statistical significance partly overlap with the last data points. Please move for clarity.

- This has been changed as requested.

2) Fig. 7E - y-axis is not in log-scale, while all other y-axes are.

- Figure 7E is shown as a linear scale because under some conditions there were zero colonies at the highest concentration of MMC, which cannot be represented in a semi-log plot.

3) Supplementary Fig. 5 appears to be missing and could not be reviewed.

- We apologize for this omission and Supplementary Figure 5 is now included.

4) The manuscript would greatly benefit from a model figure so the reader can grasp the mechanism that the authors propose.

- We agree and have now included a speculative model of SAN1 function as Supplementary figure 7. Clearly, we do not yet know every detail of the mechanism, but note that 8 years after the discovery of FAN1 its specific role in ICL repair still remains somewhat obscure.

REVIEWERS' COMMENTS:

Reviewer #1 (Remarks to the Author):

The authors have addressed the issues raised in the review.

I would suggest a minor revision to the introduction regarding Fanconi Anemia. There are now over 20 genes identified as mutant in FA. Best just to say over 20, as the number continues to change.

Reviewer #2 (Remarks to the Author):

The revised manuscript by Andrews et al. includes new work on 53BP1 foci to support the previous gammaH2AX results, additional data on the epistatic relationship between SAN1 and XPF, FAN1, and CSB (only shown in the rebuttal), respectively, as well as an RNaseH1 control for the R-loop data. Most importantly, the authors have removed the cordycepin data and repeated some of the experiments with the more specific RNA pol II inhibitor alpha-amanitin as recommended. The obtained results are inconclusive and do not support the conclusion that SAN1 acts specifically in a transcription-coupled ICL repair mechanism. Consequently, the manuscript has been rewritten in parts to avoid this conclusion, and the title was modified to emphasize the rather undefined proposed role of SAN1 in ICL repair.

Overall, the main concerns of this reviewer were addressed adequately. The presented data show convincingly that SAN1 is a 5' exonuclease with some role in the removal of ICLs, yet fundamental mechanistic aspects regarding its function and regulation remain elusive. Investigating these questions exceeds the scope of this study and will certainly be explored in the future. Removing the conclusion that SAN1 acts specifically in a transcription-coupled pathway strengthened the revised manuscript and made it more credible. The proposed model of SAN1 function (new Supplementary Fig. 7) is purely speculative, which is appropriately pointed out in the discussion. Thus, once the minor comments below have been addressed, this reviewer supports publication in Nature Communications.

Minor comments:

1) The new model figure Fig. 2F is slightly confusing and would benefit from labeling the domains and DNA termini. Also, it could be pointed out which of the shown DNA structures are bona fide substrates for SAN1 (i.e. single stranded and splayed arm structures).

2) The bars indicating statistical significance in Figs. 3, 6, and 7 were redesigned as suggested. However, the number of asterisks (i.e. the significance) in several panels have been changed while the rest of the graphs is identical. For example, in Fig. 3C (both bars have now four and previously three asterisks), in Fig. 3H (right bar has now four and previously three asterisks), in Fig. 7C (bar has now two and previously four asterisks), and in Fig. 7D (bar has now three and previously four asterisks). Have these data been reanalyzed or why has the significance been changed?

3) The nomenclature of tagged proteins throughout the manuscript is very confusing. For example, SETX seems to have both a GFP and a FLAG tag. In Fig. 7B, the input is labeled "Setx-GFP", while the pulldown is labeled "SETX-FLAG". In the text, it is mainly referred to as "SETX-FLAG-GFP". Also, the SAN1-ssf/-FLAG nomenclature is not consistent and should be improved.

4) The x-axis labels in Fig. 7J should be aligned with the data points to avoid confusion.

5) Supplementary Fig. 7B – Are the observed changes significant or not? Please indicate p values as in all other panels.

6) The order of figure panels should match the order they are mentioned in the main text. For example, panels of Supplementary Figure 3 are currently mentioned in the order a, i, j, e, b, c, d, f, g, h.

REVIEWERS' COMMENTS:

Reviewer #1 (Remarks to the Author):

The authors have addressed the issues raised in the review.

I would suggest a minor revision to the introduction regarding Fanconi Anemia. There are now over 20 genes identified as mutant in FA. Best just to say over 20, as the number continues to change.

As requested, we have changed the text in the Introduction to say >20 FA genes,

Reviewer #2 (Remarks to the Author):

The revised manuscript by Andrews et al. includes new work on 53BP1 foci to support the previous gammaH2AX results, additional data on the epistatic relationship between SAN1 and XPF, FAN1, and CSB (only shown in the rebuttal), respectively, as well as an RNaseH1 control for the R-loop data. Most importantly, the authors have removed the cordycepin data and repeated some of the experiments with the more specific RNA pol II inhibitor alpha-amanitin as recommended. The obtained results are inconclusive and do not support the conclusion that SAN1 acts specifically in a transcription-coupled ICL repair mechanism. Consequently, the manuscript has been rewritten in parts to avoid this conclusion, and the title was modified to emphasize the rather undefined proposed role of SAN1 in ICL repair.

Overall, the main concerns of this reviewer were addressed adequately. The presented data show convincingly that SAN1 is a 5' exonuclease with some role in the removal of ICLs, yet fundamental mechanistic aspects regarding its function and regulation remain elusive. Investigating these questions exceeds the scope of this study and will certainly be explored in the future. Removing the conclusion that SAN1 acts specifically in a transcription-coupled pathway strengthened the revised manuscript and made it more credible. The proposed model of SAN1 function (new Supplementary Fig. 7) is purely speculative, which is appropriately pointed out in the discussion. Thus, once the minor comments below have been addressed, this reviewer supports publication in Nature Communications.

Minor comments:

1) The new model figure Fig. 2F is slightly confusing and would benefit from labeling the domains and DNA termini. Also, it could be pointed out which of the shown DNA structures are bona fide substrates for SAN1 (i.e. single stranded and splayed arm structures).

The model has been updated and labeled in Fig. 2F to clearly display the domains and substrates, as requested.

2) The bars indicating statistical significance in Figs. 3, 6, and 7 were redesigned as suggested. However, the number of asterisks (i.e. the significance) in several panels have been changed while the rest of the graphs is identical. For example, in Fig. 3C (both bars have now four and previously three asterisks), in Fig. 3H (right bar has now four and previously three asterisks), in Fig. 7C (bar has now two and previously four asterisks), and in Fig. 7D (bar has now three and previously four asterisks). Have these data been reanalyzed or why has the significance been changed?

The change to the survival curves in Fig. 3 was an error introduced during the modification to the redesign of the figure, and should still be three asterisks as the reviewer indicates. We apologize for this mistake and it has been corrected. In Figure 6 and 7 the data were reanalyzed using a two-way ANOVA test for the entire survival curve, rather than the previous multiple comparisons test where only the significance of the highest concentration of drug was displayed.

3) The nomenclature of tagged proteins throughout the manuscript is very confusing. For example, SETX seems to have both a GFP and a FLAG tag. In Fig. 7B, the input is labeled “Setx-GFP”, while the pulldown is labeled “SETX-FLAG”. In the text, it is mainly referred to as “SETX-FLAG-GFP”. Also, the SAN1-ssf/-FLAG nomenclature is not consistent and should be improved.

In figure 7 the labels for the SETX-FLAG-GFP tagged cell line have been updated for consistency. We have also updated the SAN-ssf and SAN-FLAG tagged protein references in the text.

4) The x-axis labels in Fig. 7J should be aligned with the data points to avoid confusion.

The x-axis labels have been adjusted to be better aligned with the data points.

5) Supplementary Fig. 7B – Are the observed changes significant or not? Please indicate p values as in all other panels.

The observed changes are following alpha amanitin treatment are not statistically significant, p-values have been added to the graph as in other panels.

6) The order of figure panels should match the order they are mentioned in the main text. For example, panels of Supplementary Figure 3 are currently mentioned in the order a, i, j, e, b, c, d, f, g, h.

We have changed the order of the panels in Supplementary Fig. 3 to match the order in which they are mentioned in the text.